# HSP27 is a partner of JAK2-STAT5 and a potential therapeutic target in myelofibrosis

Margaux Sevin[1,2], Lucia Kubovcakova[3], Nicolas Pernet[1,2], Sébastien Causse [1,2], Franck Vitte[4],
Jean Luc Villeval[5,6], Catherine Lacout[5,6], Marine Cordonnier[1,2], Fernando Rodrigues-Lima[7],
Gaétan Chanteloup[1,2], Matthieu Mosca[5,6], Marie-Lorraine Chrétien[8], Jean Noël Bastie[1,2,8], Sylvain Audia[8],
Paul Sagot[8], Selim Ramla[8], Laurent Martin[8], Martin Gleave[9], Valérie Mezger[10,11,12], Radek Skoda[3], Isabelle Plo[5,6],
Carmen Garrido[1,2,13,14], François Girodon[1,2,8] & Aurélie de Thonel[1,2,10,11]

Heat shock protein 27 (HSP27/HSPB1) is a stress-inducible chaperone that facilitates cancer development by its proliferative and anti-apoptotic functions. The OGX-427 antisense oligonucleotide against HSP27 has been reported to be beneficial against idiopathic pulmonary fibrosis. Here we show that OGX-427 is effective in two murine models of thrombopoietin- and *JAKV617F*-induced myelofibrosis. OGX-427 limits disease progression and is associated with a reduction in spleen weight, in megakaryocyte expansion and, for the *JAKV617F* model, in fibrosis. HSP27 regulates the proliferation of *JAK2V617F*-positive cells and interacts directly with JAK2/STAT5. We also show that its expression is increased in both CD34$^+$ circulating progenitors and in the serum of patients with JAK2-dependent myeloproliferative neoplasms with fibrosis. Our data suggest that HSP27 plays a key role in the pathophysiology of myelofibrosis and represents a new potential therapeutic target for patients with myeloproliferative neoplasms.

[1] University of Bourgogne Franche-Comté, Dijon 21000, France. [2] INSERM, UMR 1231, Laboratory of Excellence Ligue National contre le Cancer, Dijon 21000, France. [3] Department of Biomedicine, Experimental Hematology, University Hospital Basel, Basel 20 4031, Switzerland. [4] Cypath, Dijon 21000, France. [5] University Paris XI, UMR 1170, Gustave Roussy, Villejuif 94800, France. [6] INSERM, UMR 1170, Laboratory of Excellence GR-Ex, Villejuif 94800, France. [7] Universite Paris Diderot, Sorbonne Paris Cite, Unite BFA, CNRS UMR 8251, 75013 Paris, France. [8] Hospital University Center (CHU), Dijon 21000, France. [9] The Vancouver Prostate Centre, University of British Columbia, Vancouver BC V6H 3Z6 BC, Canada. [10] CNRS, UMR7216 Épigénétique et Destin Cellulaire, Paris 75013 Cedex 13, France. [11] University of Paris Diderot, Sorbonne Paris Cité, Paris 75013 Cedex 13, France. [12] Département Hospitalo-Universitaire DHU PROTECT, Paris 75010, France. [13] Centre Georges François Leclerc (CGFL), Dijon 21000, France. [14] Scientific Cooperation Foundation (FCS) of Bourgogne Franche-Comté, LipSTIC LabEx, Dijon 21071, France. These authors jointly supervised this work: Carmen Garrido, François Girodon, Aurélie de Thonel. Correspondence and requests for materials should be addressed to F.G. (email: francois.girodon@chu-dijon.fr) or to A.d.L. (email: aurelie.dethonel@univ-paris-diderot.fr)

Heat shock proteins (HSPs) are highly conserved molecular chaperones whose synthesis is induced by different stresses in both environmental and pathophysiological conditions[1]. HSP70 (HSPA1A, NP_005336) and HSP27 (HSPB1, NP_001531 (human); Hsp27, Hspb1, NP_038588 (mouse)) are the most strongly induced proteins following aggression due to oxidative stress, anti-cancer agents or ionizing radiation for example[2]. These HSPs were originally characterized as chaperones due to their ability to prevent aggregation and restore proteostasis and to their participation in the transport of proteins. These HSPs also have a strong cytoprotective action through the inhibition of apoptosis and autophagy processes[3, 4]. Because cancer cells have to re-wire their metabolism, they need chaperones for their survival. As a consequence, HSPs are abnormally abundant in cancer cells and have become potent targets in cancer therapy[5].

HSP27 belongs to the family of small heat shock proteins and is overexpressed in solid cancers, including prostate and breast cancers, as well as in haematological malignancies[6–8]. It contributes to tumorigenesis and resistance to chemotherapy via its proliferative and anti-apoptotic functions[9, 10]. HSP27 is secreted by cancer cells and has been found in the serum of cancer patients, where its presence is often associated with a poor prognosis[11, 12]. In addition, our team recently demonstrated the beneficial effect of a new-generation antisense oligonucleotide against HSP27, OGX-427, which targets the human and rodent Hsp27 translation initiation site[13, 14], in a murine model of idiopathic pulmonary fibrosis[13]. This compound is currently under clinical evaluation (phase II clinical trials) as a chemo-sensitizing agent in solid tumours[15]. Furthermore, we and others have shown that HSP27 was strongly expressed in samples from patients with idiopathic pulmonary and kidney tubulointerstitial fibrosis[13, 16].

Myelofibrosis (MF) is a chronic degenerative disorder associated with megakaryocytic abnormalities and progressive marrow fibrosis, in which fibrous tissues replace red bone marrow. These two distinctive features are used in patients to monitor the progression of the disease[17]. The clinical features of MF generally include constitutional symptoms, splenomegaly and progressive marrow failure, which result in reduced life expectancy. MF is mostly related to myeloproliferative neoplasms (MPN) but can also be induced following treatment with haematopoietic growth factors like thrombopoietin (TPO)[17, 18]. One of the most frequent signalling pathways involved in MF pathogenesis is the Janus kinase/signal transducer and activator of transcription (JAK/STAT) pathway. The aberrant activation of the JAK/STAT pathway may result from somatic mutations directly affecting JAK activity, excessive cytokine stimulation by factors like TPO and/or epigenetic modifications leading to abnormal gene regulation[19–21].

Given the reported role of HSP27 in leukaemia and in fibrotic disorders, we hypothesized that HSP27 might be involved in MF. In this study, we show that specific inhibition of HSP27 using OGX-427 limits myelofibrosis progression in two murine models of MF[15, 22, 23] and affects the JAK2/STAT signalling pathway. Our data also reveal an increase of HSP27 in samples from patients with MF suggesting that HSP27 represents a new therapeutic target for MF.

## Results

**OGX-427 limits myelofibrosis progression in mouse models of MF.** To assess the role of HSP27 in bone marrow fibrosis, we used two animal models recommended to study the establishment of the myelofibrotic features[24]: the TPO^high and the JAK2V617F murine model (Figs. 1 and 2, respectively). These models reproduce some myelofibrotic traits found in human primary MF[22, 24], such as megakaryocyte hyperplasia, anaemia, extramedullary haematopoiesis, splenomegaly and myelofibrosis. Also, in both animal models there is sustained activation of JAK2/STAT signalling induced by the persistent production of TPO[22, 25] or the constitutive active JAK2 mutant (JAK2V617F)[26] (Figs. 1a and 2a).

As shown in Fig. 1b, we clearly observed an increase of HSP27 level in the serum in the TPO^high model compared with wild-type mice. One month later, once the myelofibrosis-like features were established, mice were treated with the specific inhibitor of HSP27, OGX-427, using a validated non-toxic dose (10 mg kg$^{-1}$)[14]. These conditions induced a 65% decrease in HSP27 expression in splenocytes from OGX-427-treated mice compared with those from control (CTL) mice (i.e., treated with a non-relevant antisense oligonucleotide, Fig. 1c). Although OGX-427 treatment in this model had no direct effects on blood parameters (Supplementary Fig. 1a), it resulted in a marked reduction in spleen weight (30%) and size, a major criterion of treatment efficacy in humans (Fig. 1d, e). In agreement with this result, spleen sections revealed hyperplasia of the red pulp in TPO^high mice and partial restoration of the white pulp territories in OGX-427-treated mice compared to CTL (Fig. 1f). These results strongly suggest that OGX-427 treatment enable partial restoration of the normal splenic architecture. In addition, the decrease in erythropoiesis in the bone marrow was more pronounced in control mice than in OGX-427-treated mice, whereas megakaryocytic hyperplasia was less marked in OGX-427-treated mice than in controls (Fig. 1g), suggesting a beneficial effect of OGX-427 treatment on erythropoiesis and on the reduction in megakaryocytic expansion. It is worth noting that in our ethically approved protocol to study the effect of OGX-427 on myelofibrosis development, the animals were killed around 5 months after the beginning of the disease. However, even though the protocol was not adequate to study the effect of OGX-427 on animal survival, the few animals that died before the planned killing suggested an improved survival in the mice treated with OGX-427 (Supplementary Fig. 1b).

We next studied the effect of OGX-427 in the JAK2V617F transgenic mouse model, which has the advantage of being a better preclinical model for fibrosis studies[26] (Fig. 2). Twelve weeks after transplantation, JAK2V617F mice were treated with OGX-427 three times per week for 10 weeks (10 mg kg$^{-1}$) or not (control) (Fig. 2a). In accordance with the results obtained in the TPO^high mice model, OGX-427 treatment significantly reduced the weight of the spleen (Fig. 2b), induced a 50% decrease in HSP27 expression in the bone marrow compared with that from control mice (Fig. 2c) and limited megakaryocyte hyperplasia in the spleen and in the bone marrow (Fig. 2d). Of note, in this model, in the OGX-427-treated mice compared with controls, we observed a decrease in reticulin fibrosis in the bone marrow sections (Fig. 2e), and this was associated with a clear fall in platelet and white blood counts (i.e., back to normal levels at the end of the treatment). In contrast, these counts remained very high in the control group (Fig. 2f).

Altogether, these results suggest a beneficial effect of OGX-427 on the pathogenesis of myelofibrosis, both by limiting splenomegaly and megakaryocyte expansion, and by reducing fibrosis development.

**Impact of HSP27 on the JAK2/STAT5 pathway in JAK2V617F-positive cells.** Since HSP27 depletion has a beneficial impact on disease progression, we explored whether HSP27 plays a role in the proliferative effect of JAK2/STAT5. HSP27 was depleted by means of two specific small interfering RNAs (siRNAs), a small

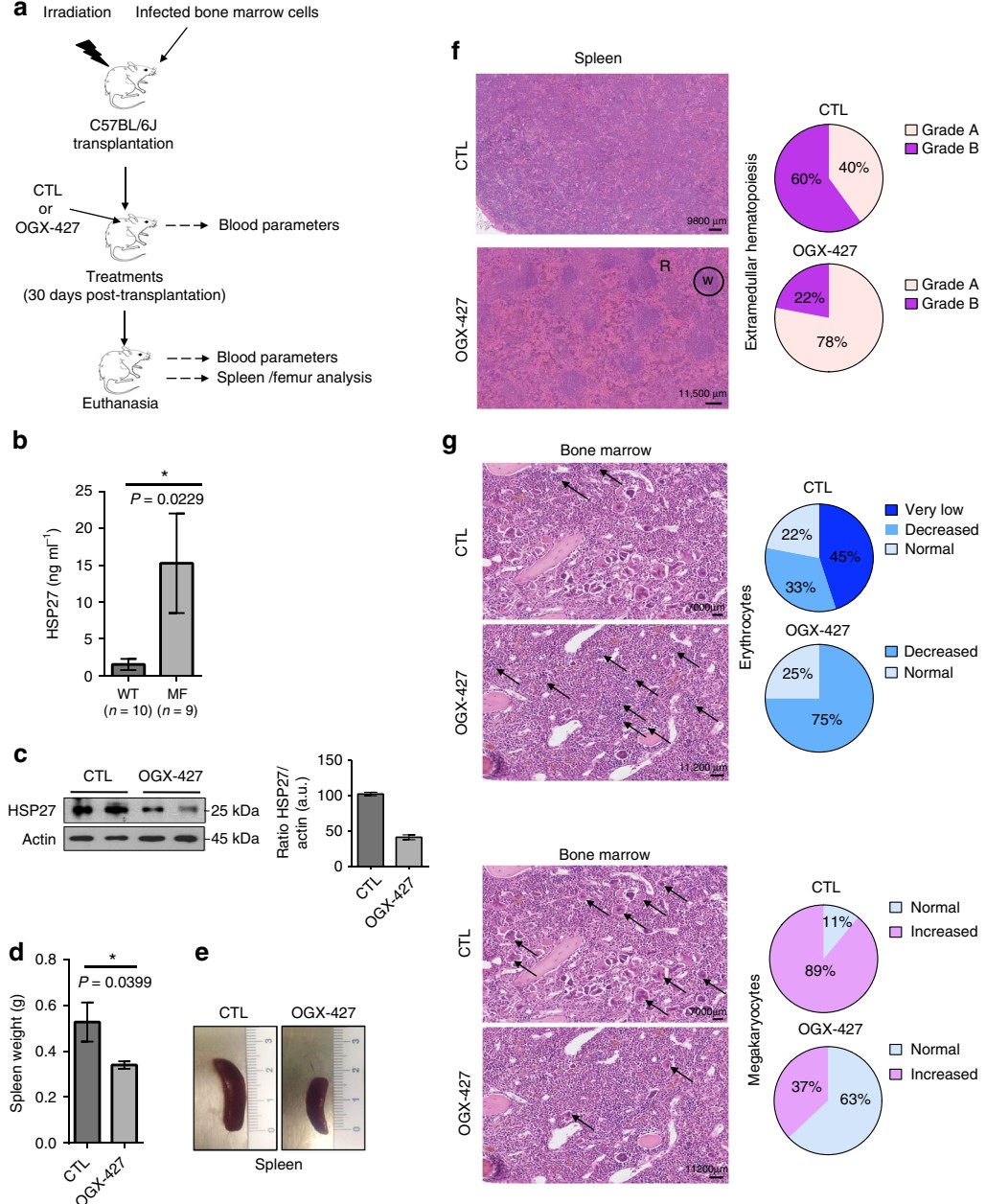

**Fig. 1** HSP27 downregulation impairs myelofibrosis progression in a TPO$^{high}$ murine model. **a** In vivo strategy of HSP27 inhibition using OGX-427, or a scrambled oligonucleotide control (CTL) injected intraperitoneally 3 times a week at a dose of 10 mg kg$^{-1}$ in a TPO$^{high}$ murine model of myelofibrosis (MF). Mice were 2–4 months old. **b** Expression level of HSP27 proteins in the serum of TPO$^{high}$ mice ($n$=9) compared with healthy mice (WT, $n$=10) measured by ELISA assay. $P$ value was calculated using the Mann–Whitney test. *$P < .05$. Error bars represent ±s.e.m. **c** Western blot analysis of HSP27 in splenocytes (whole cell lysate) of MF mice treated with OGX-427 or CTL. Actin served as the loading control. Bar graphs show quantification of mean relative amount of the proteins ($n$=2 per group). Uncropped blots presented in Supplementary Fig. 7a **d** Spleen weight was evaluated in mice ($n$=9 per group). Outcomes of the two treatments were compared using the Mann–Whitney test. *$P < .05$. Error bars represent ±s.e.m. **e** Representative picture of spleen size of MF mice treated with CTL or OGX-427. **f** Left panel, haematoxylin and eosin-stained spleen sections revealed hyperplasia of the red pulp (R) in MF mice and partial restoration of the white pulp (W) territories in OGX-427-treated mice. Right panel, assessment of the grade of extramedullary haematopoiesis (EMH), from spleen sections in 18 killed mice. Pie chart: Grade A (light pink): diffuse EMH invasion. Grade B (purple colour): diffuse EMH invasion with atrophy of white pulp. Images were obtained using a Nanozoomer scanner (Hamamatsu, France) at ×23 magnification and Calopics software. **g** Left panel, histology of bone marrow sections after haematoxylin and eosin staining. Right panel, Pie chart representing percentages of mice according to erythrocyte (right upper panel) and megakaryocyte counts (right lower panel) from bone marrow sections of 18 killed mice: very low cell number (dark blue), decreased cell number (azure blue), normal cell number (light blue) and increased cell number (pink). Arrows show erythrocyte foci and megakaryocytes. Images were obtained using a Nanozoomer scanner (Hamamatsu, France) at ×13 magnification and Calopics software

hairpin RNA (shRNA) or OGX-427 in two leukaemic cell lines, HEL92.1.7 (erythroleukaemia cell line) and SET-2 (mega-karyoblastic cell line), which bear the *JAK2V617F* mutation that constitutively activates the JAK2 signalling pathway. HSP27 depletion by the three different approaches affected cell pro-liferation induced by the constitutively activated *JAK2* mutant (Fig. 3a, Supplementary Fig. 2a, b), but not apoptosis (Supple-mentary Fig. 2c), with a more pronounced effect on the mega-karyoblastic SET-2 cell line (Fig. 3a). It is worth noting that HSP27 depletion had no effect on the proliferation of the *JAK2V617F*-negative cell line K562, suggesting that HSP27 plays

a role in *JAK2V617F*-positive cell proliferation. Our rescue experiments in the shRNA HSP27-depleted HEL92.1.7 cell line confirmed the implication of HSP27 in the cell proliferation (Supplementary Fig. 3a). An effect of HSP27 on cell proliferation was also observed on primary cells from patients by performing an erythroid colony formation assay. We found that HSP27 depletion by means of a specific shRNA induced a decrease in burst-forming units–erythroid (BFU-E) from MPN patients along with a specific effect on *JAK2V617F* cells (Supplementary Fig. 3b-d).

At the molecular level, depletion of HSP27 decreased the amount of phosphorylated STAT5 (Fig. 3b). Consequently, the

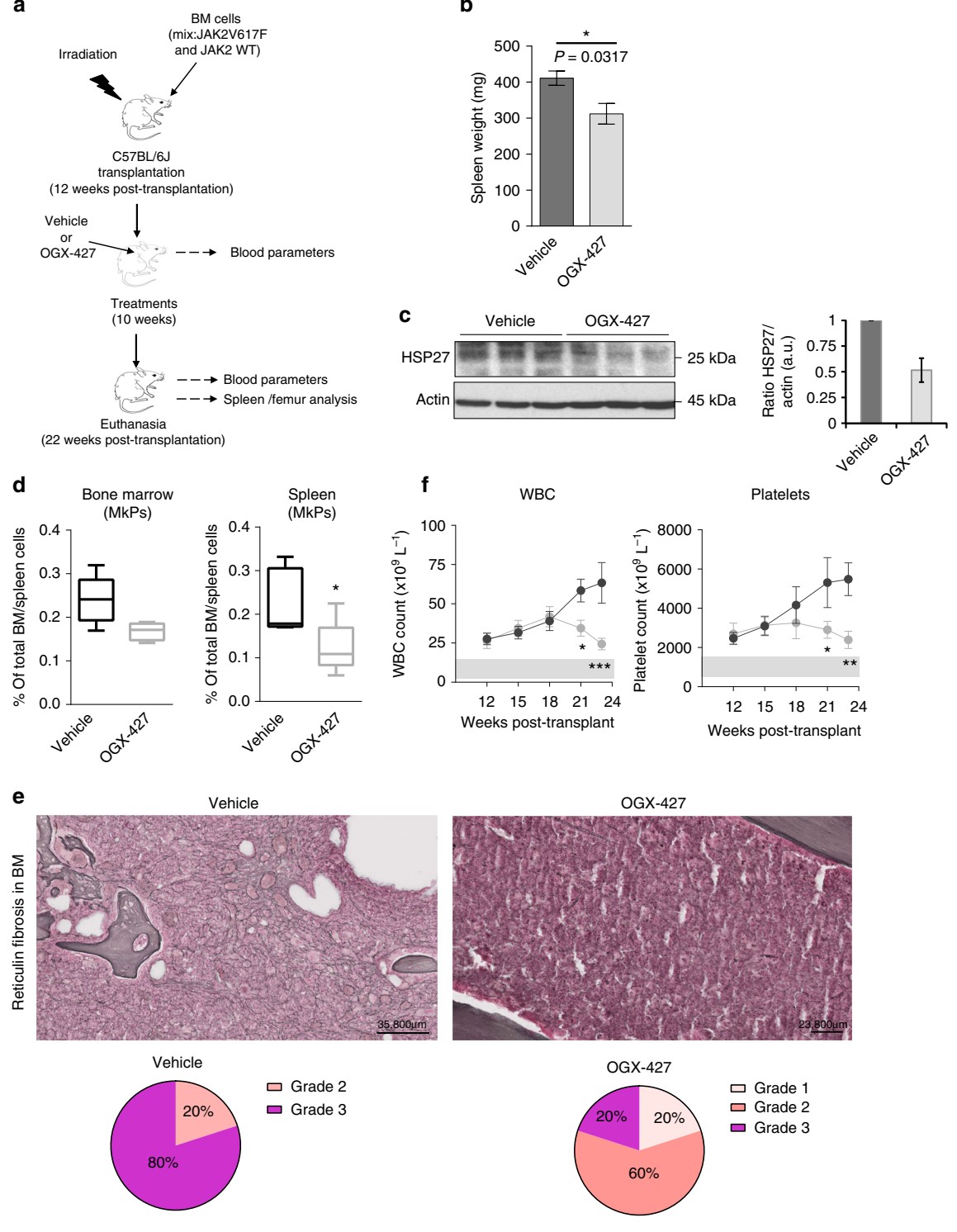

expression of the STAT5 target c-MYC, a modulator of proliferative signals, was reduced (Fig. 3b and Supplementary Fig. 2b). To investigate how HSP27 impacted phosphorylated STAT5 content, we performed both phosphorylation and de-phosphorylation in vitro assays. We found that while recombinant HSP27 did not affect the phosphorylation of STAT5 by JAK2 (Fig. 4a), it did affect its de-phosphorylation by tyrosine phosphatase SHP2 (a main phosphatase involved in STAT5 tyrosine de-phosphorylation)[27–29] (Fig. 4b). In keeping with these results, AG490, an inhibitor of JAK2 activation[30], induced a faster de-phosphorylation of STAT5 in HEL92.1.7 cells depleted for HSP27 than in control cells (Fig. 4c).

We next performed in vitro, in cellulo and in situ experiments to determine whether HSP27 interacted with JAK2/STAT5. Using biolayer interferometry, we showed that recombinant JAK2 and STAT5A/B interacted directly with immobilized biotinylated HSP27 within the same range as Crystallin alpha B (CRYAB), a well-known partner of HSP27 used here as a positive control[31] (Fig. 5a, Supplementary Fig. 4a-f).

JAK2 or STAT5 association with HSP27 was next determined by co-immunoprecipitation experiments using endogenous (Fig. 5b, c and Supplementary Fig. 5a) or exogenous tagged proteins (Supplementary Fig. 5b, c). In line with the biolayer interferometry results, HSP27 interacted with JAK2 and with STAT5.

We then confirmed these results by proximity ligation assay (PLA), by analysing the interaction between JAK2 or STAT5 and HSP27 within the HEL92.1.7 cells (Fig. 5d). The number of detected interaction foci per cell was significantly lower in siRNA HSP27 transfected than in control HEL92.1.7 cells (Fig. 5d). Using the same approach, we determined whether HSP27 affected the interaction between JAK2 and STAT5. As shown in Fig. 5d, depletion of HSP27 by siRNA significantly decreased the association between JAK2 and STAT5.

Altogether, our results indicate that the chaperone HSP27 is involved in the proliferative effect of JAK2/STAT5. HSP27, as previously shown with other HSP27 client proteins[32–34], protects STAT5 from its de-phosphorylation and might also favour the molecular assembly JAK2/STAT5.

**Patients with MPN-associated MF display high HSP27 levels.** On the basis of our previous in vivo and in vitro results, we determined the expression levels of HSP27 in patients with an MPN-associated MF. In this type of MF, acquired mutations (JAK2V617F, calreticulin (CALR) or MPL mutations) constitutively induce JAK2 signalling pathways.

As a first approach, we used flow cytometry to evaluate the HSP27 levels in CD34[+] circulating hematopoietic progenitor cells (HPCs) isolated from the blood of MF patients and healthy donors (HDs). We observed that HSP27 levels in CD34[+] HPCs from MF patients were significantly higher than those in HDs (Fig. 6a). In contrast, HSP70 and HSP90 levels within CD34[+]HPC

in MF patients were not different from those in HDs. Knowing that HSPs can also be secreted, we assessed the extracellular content of HSPs in serum from MF patients by enzyme-linked immunosorbent assay (ELISA). Again, significantly higher levels of HSP27 were detected in the serum of MF patients than in serum from HD as previously observed in our TPO[high] mouse model of myelofibrosis (Fig. 6b). High levels of extracellular HSP27 have also been reported in diseases involving chronic inflammation[35, 36], an important process in the clonal evolution of MPN[37, 38]. Notably, increased HSP27 levels in HPCs and in serum from patients were detected independently of JAK2V617F, CALR or MPL mutations (Supplementary Table 1), all three mutant proteins that are indeed known to constitutively activate the JAK2/STAT pathway. HSP70 levels in patient serum were again not different from those in HDs (Fig. 6a, b). In contrast, HSP90 levels in serum samples were higher in MF patients than in HDs, highlighting the involvement of HSP90 in MPN[39, 40] (Fig. 6b).

In line with the results described above, HSP27 was strongly expressed in bone marrow biopsies from MF patients, whereas only a few myeloid cells were positive for HSP90 and none for HSP70 (Fig. 6c). More precisely, HSP27 was detected in megakaryocytes and in endothelial cells (Fig. 6c). Previous studies also reported increased levels of HSP27 in endothelial cells from fibrotic tissues[41] and in MF megakaryocyte progenitors[42]. Altogether, these results suggest that HSP27 is a potential new player in MPN-associated MF.

## Discussion

Myelofibrosis is a severe degenerative disorder for which no curative treatment exists, except bone marrow allograft[43]. Although JAK2 inhibitors have given rise to high expectations, clinical studies, which had beneficial effects on the symptoms, demonstrated limited effects on the progression of the disease[44, 45]. Because HSP90 is known to stabilize JAK2 and therefore is a potential therapeutic target in JAK2-dependent MPN associated with MF, inhibitors of HSP90 have also been explored (currently in clinical phase II)[39, 40]. In the present work, we demonstrated that another essential chaperone, HSP27, is a therapeutic target of remarkable interest.

Inhibitors of HSPs, particularly of HSP90 and HSP27, are currently being tested in clinical trials as chemo- and radio-sensitizing agents, in many different cancer therapy protocols (reviewed in refs. [5, 46, 47]). The rationale for targeting HSPs in cancer is provided by the fact that cancer cells have a strong need in chaperones (i.e., HSPs) for their survival, because they need to rewire their metabolism. However, HSP90 inhibitors, probably due to their toxicity, induce the compensatory expression of stress-inducible HSPs (i.e., HSP70, HSP27), which, through their cell survival function, may hamper HSP90 inhibitor effectiveness[40, 48]. The HSP27 inhibitor used in this work, an antisense oligonucleotide of generation (OGX-427, in phases I and II in cancer), is a very selective inhibitor, with no or negligible

**Fig. 2** HSP27 downregulation impairs myelofibrosis progression in a JAK2V617F murine model. **a** In vivo strategy of HSP27 inhibition using OGX-427 or a vehicle, injected intraperitoneally 3 times a week at a dose of 10 mg kg$^{-1}$ in a JAK2V617F murine model of myelofibrosis (MF). Mice were 2 months old. **b** Spleen weight was evaluated in mice (n=5 per group). Outcomes of the two treatments were compared using the Mann–Whitney test. *P < .05. Error bars represent ±s.e.m. **c** Western blot analysis of HSP27 in bone marrow (whole cell lysate) of MF mice treated with OGX-427 or vehicle. Actin served as the loading control. Bar graphs show quantification of mean relative amount of the proteins (n=3 per group). Uncropped blots presented in Supplementary Fig. 7a. **d** Percentage of megakaryocytes progenitors (MkPs) present in the bone marrow or spleen, error bars represent ±s.e.m (n=5 per group). P values were calculated using the Mann–Whitney test. *P < .05. **e** Upper panel, Gordon and Sweet-stained bone marrow sections revealed reduced reticulin fibrosis in MF mice and in OGX-427-treated mice. Lower panel, assessment of the fibrosis grade from bone marrow sections of 5 mice per group. Pie chart: Grade 1 (light pink): few thin reticulin fibres. Grade 2 (light red): networks of thin reticulin fibres. Grade 3 (purple): dense network of thick reticulin fibres. Images were obtained using a Nanozoomer scanner (Hamamatsu, France) at ×23 magnification and Calopics software. **f** Blood cell parameters assessed in mice before and after OGX-427 (n=5) or vehicle treatment (n=5) and on the day of killing. WBC white blood cells. Error bars represent ±s.e.m. P values were calculated using the Mann–Whitney test. *P < .05; **P < .01. ***P < .001

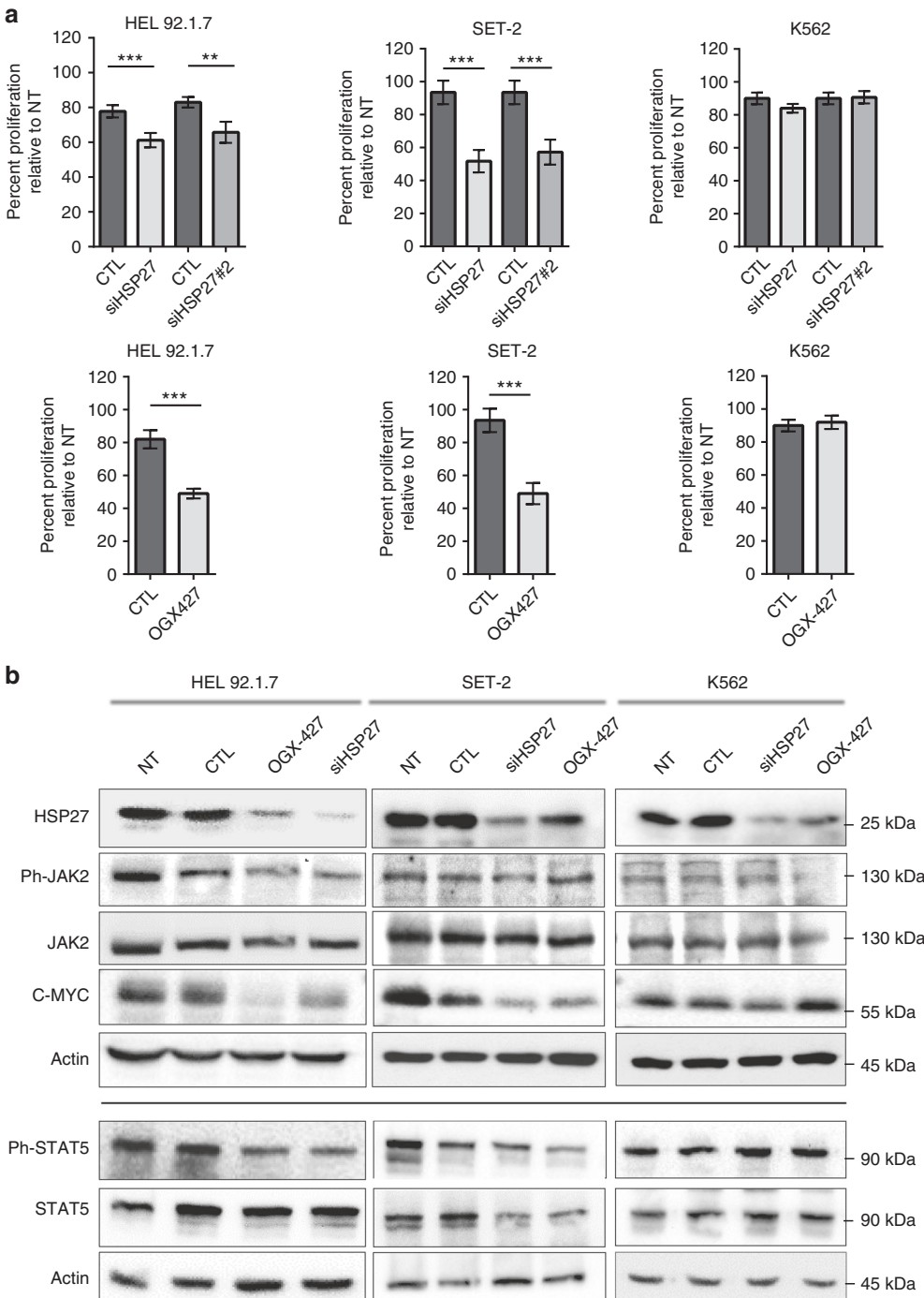

**Fig. 3** HSP27 affects proliferation of *JAK2V617F* leukaemic cell lines. **a** HEL92.1.7, SET-2 and K562 cells were transfected with HSP27 siRNA, OGX-427 or an oligonucleotide control (CTL). Bars represent cell proliferation percentages relative to non-transfected cells (NT) from 9 independent experiments. *P* values were calculated using the Mann–Whitney test. \*\**P* < .01. Error bars represent ±s.e.m. **b** HEL92.1.7, SET-2 and K562 cells were transfected with HSP27 siRNA, OGX-427 or CTL and lysed 48 h later in Laemmli Buffer. Protein expression was determined by western blot and compared to non-transfected cells (NT). Actin was used as the loading control (*n*=3 independent experiments). See quantification of the blot in Supplementary Fig. 2b. Uncropped blots presented in Supplementary Fig. 7b-f

toxicity. Moreover, in contrast to HSP90 inhibitors[49, 50], it does not induce the compensatory expression of other HSPs. Indeed, OXG-427 did not induce HSP90 or HSP70 expression in any of our models, i.e., TPO^high mice, or *JAK2V617F*-positive (HEL92.1.7 and SET-2) and -negative (K562) cell lines (Supplementary Fig. 6).

OGX-427 limited the progression of bone marrow fibrosis in the *JAK2V617F* model, but not in the TPO^high model (Fig. 2e). In

accordance with anterior published work, Fedratinib (a selective JAK2 inhibitor for which clinical studies have been hindered because of its toxicity) also decreased splenomegaly without affecting fibrosis in this TPO^high murine model, whereas it impaired fibrosis in the post-polycythemia vera myelofibrosis murine model[51, 52]. The differences in results obtained, using different mouse models, might be explained by the fact that the process of fibrosis in the TPO^high model is very rapid and

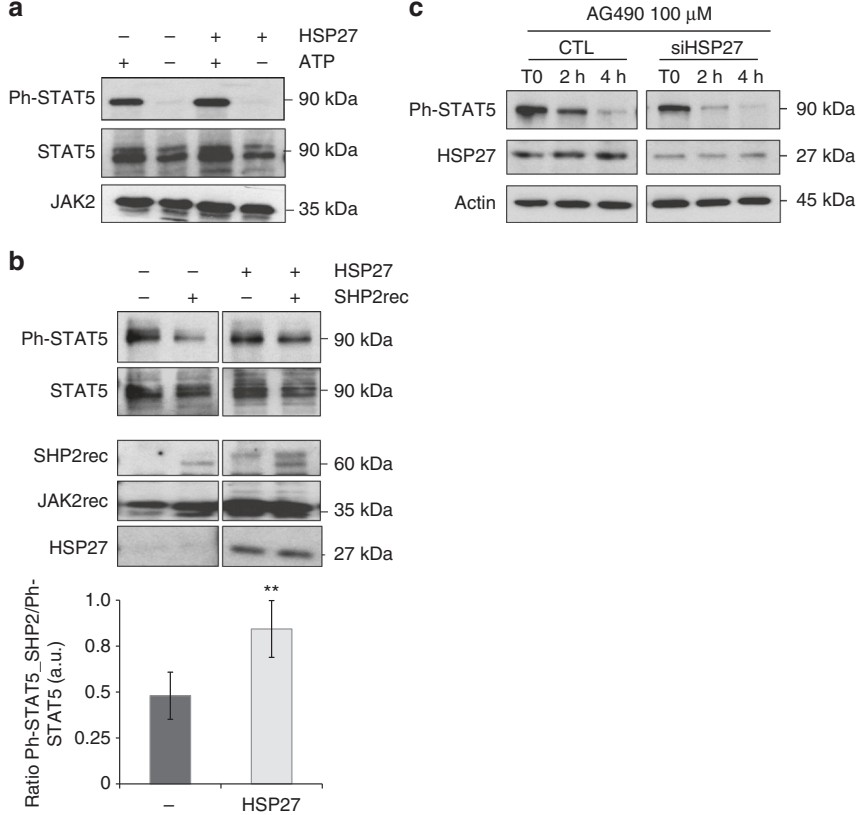

**Fig. 4** HSP27 affects de-phosphorylation of STAT5. **a** The phosphorylation of STAT5 by JAK2 is HSP27 independent. An in vitro kinase assay was performed using the recombinant protein JAK2 and STAT5 (produced in reticulocyte lysate) in the presence or absence of HSP27 (produced in reticulocyte lysate). The level of STAT5 phosphorylation, and the amount of JAK2 and STAT5 proteins were determined by western blot ($n$=3 independent experiments). Uncropped blots presented in Supplementary Fig. 8a. **b** HSP27 impairs the in vitro de-phosphorylation of STAT5 induced by SHP2. Following an in vitro kinase assay using the recombinant protein JAK2 and STAT5 (same batch produced in reticulocyte lysate), samples were incubated or not with the recombinant phosphatase, SHP2, in the presence or absence of HSP27 (produced in reticulocyte lysate). The level of phosphorylated STAT5 was detected by western blot using an antibody specific for Ph-STAT5. The expressions of JAK2, SHP2 and HSP27 are also shown. Bar graphs show normalized ratios of Ph-STAT5_SHP2/Ph-STAT5 bands quantified from the western blots ($n$=4 independent experiments), error bars represent the ±s.d. P values were calculated using the Student's $t$ test. **P < .01. Uncropped blots presented in Supplementary Fig. 8b. **c** HEL92.1.7 cells were transfected with HSP27 siRNA or CTL for 48 h. Then, cells were treated or not with the JAK2 inhibitor, AG490 (100 μM), at indicated times and lysed in Laemmli Buffer. Level of STAT5 phosphorylation was determined by western blot. Actin was used as the loading control ($n$=3 independent experiments). Uncropped blots presented in Supplementary Fig. 8c

irreversible[22, 25] compared to other models. Despite the limits of this TPO^high model of MF, we nonetheless succeeded to demonstrate the effectiveness of OGX-427, not only on splenomegaly, but also on extramedullar hematopoiesis. This suggests a potential implication of HSP27 in the migration of bone marrow stem cells, from the medullary niche to extramedullary foci.

It has been shown that HSP27 is overexpressed in fibrotic conditions such as pulmonary or renal fibrosis[16, 53, 54]. For instance, we previously unravelled the overexpression of HSP27 in the pleura of patients with pulmonary fibrosis, as well as in two rat models of idiopathic pulmonary fibrosis[13]. The identification, in the present work, of HSP27 overexpression in MF patients (Fig. 6a–c), as well as in mouse models of MF (Figs. 1 and 2), again supports association between HSP27 and fibrotic pathologies. How HSP27 impacts myelofibrosis development is not yet understood, but it may involve the JAK2/STAT5 signalling pathway. Indeed, our results demonstrate that HSP27 interacts directly with JAK2 and STAT5 (Fig. 5a–d), probably favouring their assembly into a stable complex, and protecting STAT5 from its de-phosphorylation (Fig. 4b, c). Additionally, we show that HSP27 is involved in the proliferation of *JAK2V617F* cells and the

expression of c-MYC (Fig. 3b), a downstream target of STAT5 and a known actor in fibrosis development[55, 56]. Interestingly, the HSP70/90 co-chaperon STIP1 (stress-induced phosphoprotein 1) has been reported to be necessary for the control of the stability of the JAK2/HSP90/STAT3 complex[57], suggesting that JAK2/STAT5 activity is tightly regulated by different chaperones. Other signalling pathways might also be involved in the myelofibrotic process. In particular, it has been reported that stimulation of platelets by collagen leads to a HSP27-dependent release of platelet-derived growth factor (PDGF), a growth factor that participates in the pathogenesis of MF[58]. In addition, aberrant expression of PDGF and its receptor (PDGF receptor alpha) has been identified in bone marrow cells of patients with idiopathic MF at the advanced stages of the disease[59].

Here, we also identify for the first time an extracellular increase of both HSP90 and HSP27 in the sera of MF patients (Fig. 6b), as well as of mice developing the disease (Fig. 1b). Although HSP90 extracellular function in MF remains to be studied, in cancer it has been reported to participate in carcinogenesis[60, 61]. In the context of pathological remodelling heart, García et al.[62] have shown that inhibition of extracellular HSP90 lessened the yield of

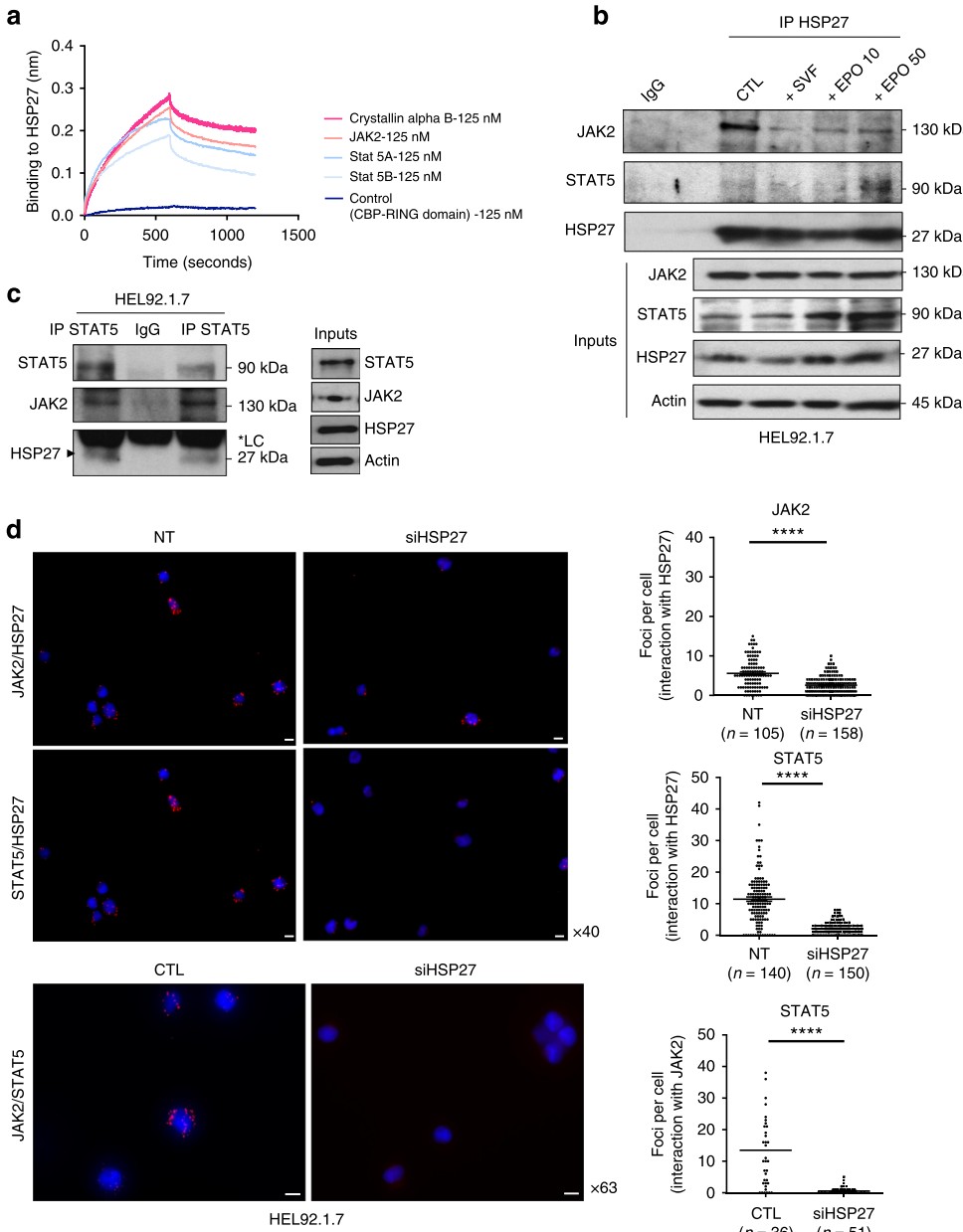

**Fig. 5** HSP27 interacts with JAK2/STAT5. **a** The binding of recombinant JAK2 and STAT5a/b at 125 nM to immobilized biotinylated HSP27 was determined by biolayer interferometry Crystallin alpha B and the CBP RING domain serves as a positive control and negative control, respectively (Supplementary Fig. 4a-c) (*n*=3 independent experiments). **b** Immunoprecipitation from HEL92.1.7 cell extracts of endogenous HSP27 was followed by immunodetection of endogenous STAT5 and JAK2. CTL: unstimulated cells starved for 16 h; +SVF: Cells starved for 16 h and then stimulated with SVF (10%) for 30 min. Inputs: proteins in total cell lysates. IP IgG: immunoprecipitation with a non-relevant antibody (IgG mouse) (*n*=2 independent experiments). Uncropped blots presented in Supplementary Fig. 8d. **c** Immunoprecipitation of endogenous STAT5 was followed by immunodetection of endogenous HSP27 and JAK2. Inputs: proteins in total cell lysates. IP IgG: immunoprecipitation with a non-relevant antibody (IgG mouse) (*n*=2 independent experiments). Uncropped blots presented in Supplementary Fig. 8e. **d** Immunofluorescence analysis of the endogenous interaction (red foci) of JAK2 (upper panel) or STAT5 (middle panel) with HSP27 visualized in situ by PLA in HEL92.1.7 cells transfected or not (NT) with a HSP27 siRNA. Nuclei are stained with DAPI. Images were taken randomly and obtained using an Axio Imager 2 at ×40 magnification and analysed using ICY software. Cells were segmented manually, and the number of interaction foci in each cell was counted using the spot detector plugin. Right panel, graphs represent quantification of the interaction of JAK2 or STAT5 with HSP27 visualized in situ by PLA. Each data point corresponds to an analysed cell, placed according to the number of detected interaction foci (lower panel). Immunofluorescence analysis of the endogenous interaction (red foci) of JAK2 with STAT5 visualized in situ by PLA in HEL92.1.7 cells transfected with a scramble (CTL) or a HSP27 siRNA (siHSP27). Nuclei are stained with DAPI. Images were obtained using an Axio Imager 2 at ×63 magnification and analysed using ICY software as in upper panel. Right panel, graphs represent quantification of the interaction of endogenous JAK2 with STAT5 visualized by PLA. P values were calculated using the Mann–Whitney test. ****P < .0001. Scale bar 10 μm

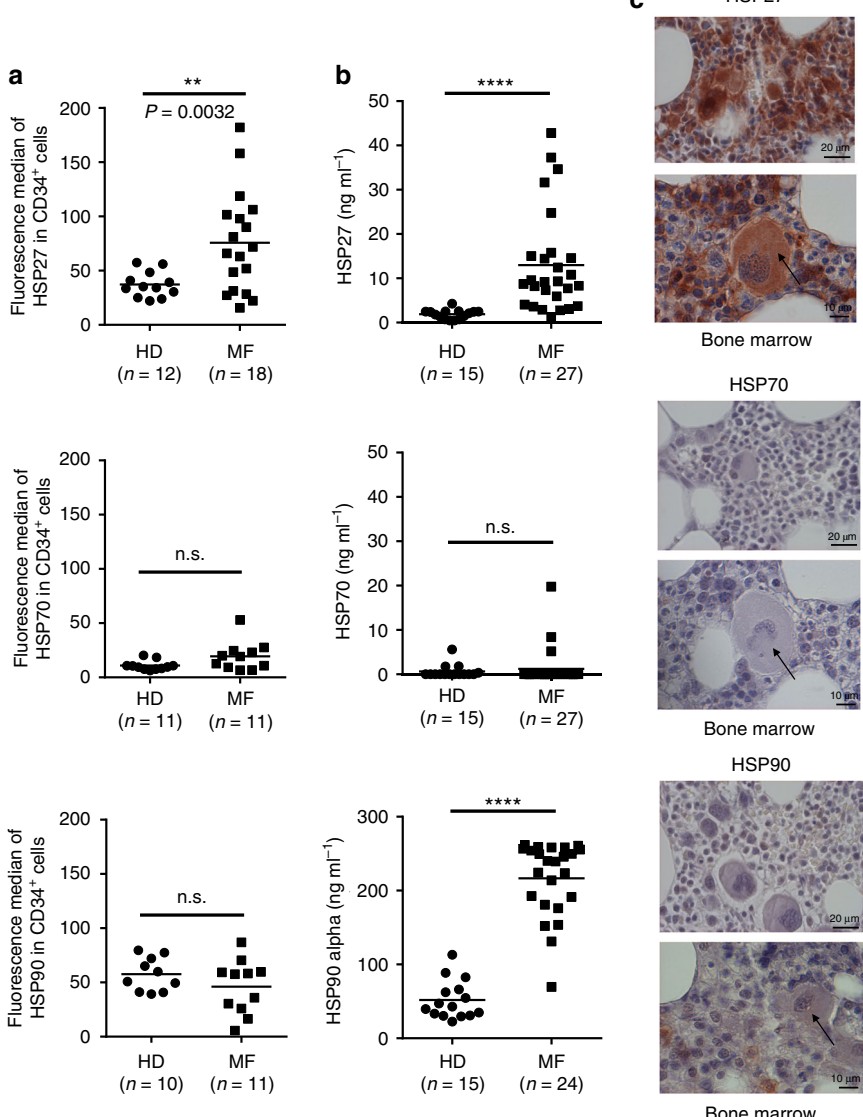

**Fig. 6** HSP27 is overexpressed in patients with MPN-associated MF. **a** Flow cytometry analysis of HSP27, HSP70 and HSP90 in circulating hematopoietic progenitor CD34+ cells from patients with myelofibrosis (MF) and healthy control donors (HDs) (number of samples analysed; MF: $n$=11–18; HD: $n$=10–12). $P$ values were calculated by using the unpaired $t$ test with Welch's correction. **$P$ < .01; n.s. not significant. Expression levels of HSP27, HSP70 and HSP90 from MF patients were plotted as median fluorescence intensities. **b** Analysis of HSP27, HSP70 and HSP90 levels in the serum of patients with MF ($n$=24-27) compared with HDs ($n$=15) measured by ELISA. $P$ values were calculated using the Mann–Whitney test. ****$P$ < .0001. **c** Representative histological sections of bone marrow from MF patients. Images were obtained using Cell Observer (Zeiss, France) at two different magnifications (×23, scale bar 20 μm and ×63, scale bar 10 μm) and Axiovision software. Arrows show megakaryocytes

collagen production as well as the canonical transforming growth factor β (TGF-β) signalling cascade. It has been demonstrated that extracellular HSP90 binds to the extracellular part of TGFβ receptor I (TGFβRI) in cardiac fibroblasts, thereby inducing the canonical pathway of collagen production. Accordingly, *Hsp*90 knockout mice showed a dramatic decrease in the production of collagen by fibroblasts in response to TGF-β stimulation[62]. These results highlight the existence of a functional cooperative partnership between HSP90 and TGFβRI in the fibrotic process. Since we also observed a marked increase in serum HSP90 levels in MF patients, this suggests that extracellular HSP90 might likewise play a role in the pathogenesis and/or could be used as a biomarker in MF[63].

Extracellular increase of HSP27 has already been observed in pathological conditions, such as chronic pancreatitis, pancreatic cancer or breast cancer[64–66]. However, in pulmonary and renal

fibrosis, in which a significant increase of intracellular HSP27 was previously reported, its levels in the patient's sera have not yet been investigated[16, 53, 54]. The level of HSP27 in the extracellular medium is generally correlated with an increase in its intracellular concentration[67], which is observed under extreme conditions such as very intense physical effort[68] or pathological secretion by cancer cells[11, 69, 70]. Accordingly, we show, here, that the high level of HSP27 in the serum of MF patients, compared to healthy donors, correlated with an increase in HSP27 intracellular levels in CD34+, megakaryocytes and endothelial cells (Fig. 6a, c). We believe that HSP27 secretion by these cells contributes to the high levels of HSP27 detected in the blood of MF patients, although we do not exclude that other types of cells may also participate to this process. In particular, it has already been shown that HSP27 can be released by human platelets under normal conditions[71]. Platelets might therefore also be involved in HSP27 secretion in the

context of MF, especially as they are deregulated in this disease[42]. Altogether, the increase in the expression of HSP27 observed in CD34+ HPC, as well as in the serum from MF-associated MPN patients, suggests that HSP27 is a potential therapeutic target in this disease.

To summarize, using two murine models of myelofibrosis, we show that a specific HSP27 inhibitor, OGX-427, limits the progression of myelofibrosis by (i) reducing both spleen weight and size, (ii) decreasing myeloid proliferation in the spleen and bone marrow, and by reducing megakaryocytic expansion, (iii) decreasing reticulin fibrosis and (iv) normalizing the platelet and white blood counts. We also show, for the first time, that HSP27 may chaperone JAK2/STAT5, a central signalling pathway activated in MF-associated MPN, and also in other kinds of diseases[72,73]. Altogether, our results support a key role for HSP27 in the pathophysiology of MF and highlight a potential interest of HSP27 inhibitors as a complementary approach for MF-associated MPN treatment.

## Methods

**Reagents**. OGX-427, a second-generation antisense oligonucleotide inhibitor (Patent PCT no. 10/605, 498, 2005; http://ir.oncogenex.com/) and the oligonucleotide control were synthetized by OncoGenex Pharmaceuticals. AG490 was purchased from Invivogen (100 μM). Antibodies used for western blotting included anti-JAK2 (dilution 1:1000, clone D2E12, #3230), anti-phospho-JAK2 (dilution 1:1000, polyclonal, #3771), anti-STAT5 (dilution 1:1000, clone D2O6Y, #9363), anti-phospho-STAT5 (dilution 1:1000, clone D47E7, #4322) and anti-c-MYC (dilution 1:1000, polyclonal, #9402) purchased from Cell Signalling Technologies; anti-HSP27 (dilution 1:1000, polyclonal, SPA-803 (human); dilution 1:1000, clone G3.1, SPA801 (murine)), -HSP70 (dilution 1:1000, clone C92F3A-5, SPA-810) and -HSP90 (dilution 1:1000, clone AC88, SPA-830) were purchased from Enzo Life Sciences and anti-beta-actin-peroxidase (dilution 1:25000, clone AC15, A3854) from Sigma-Aldrich. Antibodies used for PLA included anti-HSP27 (dilution 1:100, clone G3.1, SPA-800); anti-JAK2 (dilution 1:100, clone D2E12, #3230); and anti-STAT5 (dilution 1:100, clone D2O6Y, #9363). Antibodies for flow cytometry included anti-CD34 PE-Cy7 (dilution 5:100, clone 8G12, #348811), -CD45 APC (dilution 20:100, clone HI30, #555485), -Mouse IgG FITC purchased from BD Biosciences and anti-HSP27 FITC (dilution 1:200, clone G3.1, SPA-800FI), -HSP70 FITC (SPA-810FI, clone C92F3A-5, dilution 1:100), -HSP90 Alexa 488 (dilution 1:10, clone AC88, SPA830-488) from Enzo Life Sciences. Secondary antibodies were purchased from Jackson Immunoresearch Laboratories. The two siRNAs against HSP27 (siRNA#1: 122371, "GUUCAAAGCAACCACCUGUtt", siRNA#2: S6991, "GCGUGUCCCUGGAUGUCAAtt") were purchased from Ambion and the siRNA control (Negative control #1) from Sigma-Aldrich.

**Cell lines and transfections**. The *JAK2V617F*-positive human leukaemic cell line HEL92.1.7 (ACC-11, from DSMZ) and SET-2 (ACC-608, from DSMZ), the *JAK2V617F*-negative human leukaemic cell line K562 (ACC-10, from DSMZ) and the human embryonic kidney (HEK293T) cell line (ATCC CRL-3216) were cultured in RPMI-1640 medium supplemented with 10% (HEL92.1.7, K562 and HEK293T) or 20% (SET-2) v/v foetal bovine serum. The cell line GP+E-86 (ATCC CRL-9642) was cultured in Dulbecco's modified Eagle's medium (DMEM) supplemented with 10% v/v foetal bovine serum. All cell lines were tested for mycoplasmas. HEL92.1.7, SET-2 and K562 cells were transfected with siRNA against HSP27 (siHSP27), siRNA control (CTL) or OGX-427 using Amaxa Nucleofector system (Lonza) according to the suppliers' instructions.

**Proliferation assay**. Non-transfected (NT) cells and cells transfected with siHSP27, OGX-427 and siRNA control (CTL) were plated in 96-well plates in triplicate, 48 h post transfection, in 100 μl of medium. Cell proliferation was determined by the cell proliferation assay (XTT, Roche). Briefly, 50 μl of XTT labelling mixture was added in each well for 4 h and cell proliferation was determined by absorption at 490 nm with a reference wavelength at 650 nm.

**Western blot analysis**. Protein extracts from cells and spleen tissues were prepared using a modified Laemmli buffer (5% sodium dodecyl sulphate, 10% glycerol, 32.9 mM Tris-HCl pH 6.8) supplemented with protease and phosphatase inhibitors (Roche). Then, 30 μg of proteins from each lysate were subjected to migration on 8–12% acrylamide gels and transferred to polyvinylidene difluoride membranes (GE Healthcare Europe GmbH) in borate buffer (50 mM Tris-HCl and 50 mM borate) for 1 h and 45 min at constant voltage (48 V). The membranes were incubated with primary antibodies overnight at 4 °C, then washed in Tris-buffered saline–Tween 0.1% and incubated for 1 h with horseradish peroxidase (HRP)-coupled secondary antibody or beta-actin peroxidase antibody (Sigma-Aldrich). The signal was revealed using a chemiluminescent reagent (Luminata Crescendo

Western HRP substrate reagent, Merck Millipore) and was detected using a ChemiDoc XRS System (Bio-Rad). Uncropped blots are presented in Supplementary Fig. 7, 8.

**Protein interaction studies**. For in vitro interaction experiments, we used biolayer interferometry technology (Octet Red, Forté-Bio). Recombinant HSP27 (ADI-ESP-715, Enzo-Life Sciences (produced in *Escherichia coli*, low endotoxin)) was desalted (Zeba™ Spin Desalting Columns, 7K molecular-weight cutoff, 0.5 ml (1034–1164, Fisher Scientific)) and biotinylated at a molar ratio biotin/protein (3:1) for 30 min at room temperature (EZ-Link NHS-PEG4-Biotin (1189-1195, Fisher Scientific)). Excess Biotin was removed using Zeba™ Spin Desalting Columns. Biotinylated recombinant HSP27 was used as a ligand and immobilized at 10 μg ml$^{-1}$ on streptavidin biosensors after dilution in phosphate-buffered saline (PBS; 600 s). Interactions with desalted analytes diluted in PBS at 125 nM (recombinant STAT5a/b (TP305753 and TP309429 respectively, Origen), JAK2 (TP320503, Origen) or the RING domain of CBP (gift from Dr. Rodrigues-Lima used as a negative control[74] or crystallin alpha B as a positive control (ADI-SPP-228, Enzo-Life Sciences)) were analysed after association (600 s) and dissociation (600 s) steps at 26 °C. The specificity of the BLI experiments was tested and validated as shown in Supplementary Fig. 4d-f.

For in vitro immunoprecipitation, recombinant proteins were produced using TNT Quick Coupled Transcription/Transcription System (L1170, Promega) as follows: 1 mg of template plasmid DNA was added to the reaction mixture, which was afterwards incubated at 30 °C for 90 min. The in vitro translated proteins were incubated for 15 min at 37 °C in buffer (Hepes pH 7.4 10 mM, NaCl 25 mM, MgCl$_2$ 5 mM, MnCl$_2$ 5 mM, dithiothreitol (DTT) 1 mM, protease and phosphatase inhibitors) and then subjected to immunoprecipitation for 1 h and 30 min at RT using an anti-mouse HSP27 antibody (2 μg, clone G3.1, SPA-800, Enzo Life Sciences) or a mouse IgG (serum, I5381, Sigma-Aldrich) as a negative control that were pre-incubated 1h at RT with Protein G UltraLink Resin (53132, Pierce). Protein complexes were then washed 4 times in lysis buffer and suspended in 2X laemmli buffer. After boiling, the immunoprecipitates were resolved in 8-12% SDS-PAGE and immunoblots were performed using an anti-rabbit STAT5 (#9363, Cell Signalling technologies).

For endogenous interaction experiments, we performed co-immunoprecipitations and proximity ligation assays (PLA) in HEL92.1.7 cells. For co-immunoprecipitation, where indicated, HEL92.1.7 cells were first deprived of serum for 16h and then stimulated for 10 min with 10% serum or increasing doses of erythropoietin (EPO, 15 and 50 UI per ml). Cells were lysed in Lysis buffer (50 mM Hepes pH8, 150 mM NaCl, 5 mM EDTA, Triton X-100 0.1%, Glycerol 10%, 0.25 mM DTT, NaVO$_4$ 10 mM, NaF 10 mM, proteases inhibitors). Precleared cell lysates were subjected to immunoprecipitation for 3 h using an anti-mouse HSP27 antibody (2 μg, SPA-800, Enzo Life Sciences) or anti-STAT5 mouse (4 μg, clone A-9, sc-74442, Santa-Cruz) or a mouse IgG (I5381, Sigma-Aldrich) as a negative control that had been pre-incubated for 1 h at room temperature (RT) with Protein G UltraLink Resin beads (53132, Pierce). Protein complexes were then washed 4 times in lysis buffer and suspended in 2× laemmli buffer. After boiling, the immunoprecipitates were resolved in 8–12% sodium dodecyl sulphate–polyacrylamide gel electrophoresis (SDS-PAGE) and immunoblots were performed using an anti-rabbit STAT5, JAK2 or HSP27 antibody. For PLA, HEL92.1.7 cells transfected or not with HSP27 siRNA#1 or control siRNA were fixed in 4% paraformaldehyde, permeabilized in chilled methanol, blocked with 3% bovine serum albumin, then incubated overnight with anti-HSP27 and anti-JAK2 or -STAT5 antibodies. PLA was carried out according to the supplier's instructions (Olink). Cells were mounted with 4,6-diamidino-2-phenylindole (DAPI) containing prolong diamond (Life Technologies). Interactions between JAK2 or STAT5 and HSP27 were visualized in situ using an axio imager 2 (Zeiss, France) and analysed using ICY software.

For endogenous immunoprecipitation, SET-2 cells were lysed in Lysis buffer (50 mM Hepes pH 8, 150 mM NaCl, 5 mM EDTA, Triton X-100 0,1%, Glycerol 10 %, 0.25 mM DTT, NaVO$_4$ 10 mM, NaF 10 mM, proteases inhibitors). Precleared cell lysates were subjected to immunoprecipitation overnight using an anti-STAT5 mouse (4 μg, clone A-9, sc-74442, Santa-Cruz) or an anti-HSP27 rabbit (3 μg, polyclonal, SPA-803, Enzo Life Sciences) with a mouse or a rabbit IgG (I5381/I5006, Sigma-Aldrich), respectively, as a negative control that were pre-incubated 1 h at RT with Protein G UltraLink Resin beads (53132, Pierce). Protein complexes were then washed 4 times in lysis buffer and suspended in 2× laemmli buffer. After boiling, the immunoprecipitates were resolved in 10% SDS-PAGE and immunoblots were performed using an anti-rabbit STAT5, JAK2 (clone D2O6Y, #9363 and clone D2E12, #3230, Cell Signalling technologies) or HSP27 antibody (polyclonal, SPA-803, Enzo Life Sciences).

For exogenous immunoprecipitation, HEK293T cells were transfected with HA-tagged HSP27 and Myc-tagged STAT5a or JAK2V617F constructs. After 24 h, cells were lysed in Lysis buffer (50 mM Hepes pH8, 150 mM NaCl, 5 mM EDTA, Triton X-100 0.1%, Glycerol 10%, 0.25 mM DTT, NaVO$_4$ 10 mM, NaF 10 mM, proteases inhibitors). Precleared cell lysates were then subjected to immunoprecipitation for 3 h using an anti-mouse HA-tag (HSP27) (clone H-7, H3663, Sigma-Aldrich) or anti-mouse Myc-tag (clone 9B11, #2276, Cell Signalling technologies), STAT5 antibody (clone A-9, sc-74442, Santa-Cruz) or a mouse IgG as a negative control (serum, I5381, Sigma-Aldrich) that were pre-incubated for 1 h at RT with Protein

G UltraLink Resin beads (53132, Pierce). Protein complexes were then washed 4 times in lysis buffer and suspended in 2× Laemmli buffer. After boiling, the immunoprecipitates were resolved in 8–12% SDS-PAGE and immunoblots were performed using an anti-rabbit STAT5, JAK2 or HSP27 antibody.

**ShRNA production and cell infection.** For shRNA experiments, the sequence for shRNA targeting human HSP27 (forward-gatcccc (GATCACCATCCCAGT-CACCTT, sens) ttcaagaga (AAGGTGACTGGGATGGTGATC, antisens) tttttggaaa and reverse-agcttttccaaaaa (GATCACCATCCCAGTCACCTT, sens) tctcttgaa (AAGGTGACTGGGATGGTGATC, antisens) ggg) (Eurofins Genomics) was cloned in the pRRLsin-PGK-eGFP-WPRE vector (Addgene, #12252) using a shuttle vector pH1 and the ligase T4 DNA (M0202, New England Biolabs). Ligation products were transformed in XL10-Gold® Ultracompetent Cells (50-125-094, Aligent Technologies). Bacteria were incubated at 32 °C overnight. PRRL-shRNA-HSP27 final product was checked by sequencing (Genewiz, Takelay, UK). Lentivirus particles were produced in HEK293T cells after transfection of the cells with the plasmids PRRL-HSP27 (15 μg), psPAX2 (10 μg, 122160, Addgene) and pMD2. G (5 μg, 12259, Addgene) using the calcium chloride transfection method. Viral particles were concentrated by ultracentrifugation (45 min, 20,000 × $g$). HEL92.1.7 cells were then transduced with lentivirus particles (1.2.10$^6$ U per 500,000 cells) encoding shRNA for HSP27 and sorted for green fluorescent protein (GFP) on ARIA III cytometer.

**Phosphorylation and de-phosphorylation assays.** For the phosphorylation assay, STAT5 (produced in reticulocyte lysate) was incubated for 15 min at 30 °C in the presence or absence of HSP27 (produced in reticulocyte lysate) with JAK2 recombinant kinase (50 ng, 14-640, Merck Millipore) in a 15 μl of assay kinase buffer (Hepes pH 7.4, 10 mM, NaCl 50 mM, MgCl$_2$ 5 mM, MnCl$_2$ 5 mM, DTT 1 mM, NaF 10 mM) in the presence or absence of ATP (250 μM, #9804, Cell Signalling Technology) and the reaction was stopped by adding 2× Laemmli buffer.

In vitro transcription and translation reactions were performed using TNT Quick Coupled Transcription/Transcription System (L1170, Promega) as recommended by the supplier. Next, 1 μg of the plasmid DNA template was transcribed and the protein was translated at 30°C for 90 min.

For the de-phosphorylation assay, a phosphorylation kinase assay JAK2/STAT5 was first done in the presence of ATP as described above and the reaction was stopped by adding 20 μM of Ruxolitinib (sc-396768A, Santa-Cruz) for 5 min. Then, HSP27 (produced in reticulocyte lysate) was added or not to the samples and incubated for 10 min at 30 °C. The recombinant SHP2 (590 ng, SRP0217, Sigma-Aldrich) was added or not to the samples for 12 min at 30 °C and the reaction was stopped by adding 2× Laemmli buffer.

**Murine models and analysis of mice.** For the TPO$^{high}$ murine model, specific pathogen-free C57BL/6J female mice between 2 and 4 months of age were purchased from Janvier (Lyon, France) and maintained in sterile housing in accordance with the guidelines of the Ministère de la Recherche et de la Technologie (Paris, France). Rodent laboratory food and water were provided ad libitum. All experiments were granted by the ethics committee CE26 (Paris, license D94-076-11SCEA ; IGR 23-11-12 2012-061). All animal procedures were performed with inhalation anaesthesia with isoflurane (TEM, Lormont, France). To generate the TPO$^{high}$ model, mice were lethally irradiated (9.5 Gy, 60 Co gamma rays) and animals were injected intravenously (IV) with 1 to 5 × 10$^6$ transduced bone marrow cells (retrovirus MPZenTPO)[22, 25]. The infection of bone marrow cells was done as follow: 4 days after 5-fluorouracil injection in mice (150 mg kg$^{−1}$ administered IV), bone marrow cells (from two femurs) were co-cultivated with 1 × 10$^5$ MPZenTPO-producing GP/E-86 cells in media containing DMEM (20 ml), 20% foetal calf serum (FCS), 10% pokeweed mitogen-stimulated spleen cell conditioned medium (PWM-SCM), in a 100 mm tissue culture petri dish. After 4–5 days, nonadherent cells were harvested and injected into lethally irradiated mice. The mice were randomized based on blood count 30 days after transplantation and injected intraperitoneally, three times per week, with 10 mg kg$^{−1}$ OGX-427, or a control oligonucleotide. TPO$^{high}$ mice were treated for 5 weeks and euthanized 4 weeks after the last injection (Fig. 1a).

For the *JAK2V617F* murine model, bone marrow cells (2 × 10$^6$ per recipient) from 2-month-old female C57BL/6J mice (control mice, Harlan Laboratories) and SclCre;FF1;GFP mice were transplanted into 2-month-old female C57/BL6J lethally irradiated recipients (5 mice per group) in a 1:1 ratio[26]. At 12 weeks post transplantation, a first blood cell count was done to check and confirm the MPN phenotype. The animals were then divided into two groups of five mice: one group was treated with the vehicle alone (3 times per week) and the other mice were treated with the HSP27 inhibitor OGX-427 (10 mg kg$^{−1}$; 3 times per week). The mice were treated for 10 weeks and cell blood count was performed every 3 weeks. At the end of the experiment, the mice were killed, and spleens were weighed and stored at −20 °C. The female mice had free access to food and water and were kept under specific pathogen-free conditions (laboratory animal husbandry license CH-1007H). Animal experiments were done in strict compliance with Swiss laws for animal welfare and were authorized by the Swiss Cantonal Veterinary Office of Basel-Stadt, license 2581.

For the two models, haematocrit, platelet and white blood cell (WBC) counts were determined using an automated counter (Scil Vet abc Plus+, Horiba) on

blood collected from the tails in EDTA tubes throughout the study. At the end of the study, spleens were weighed, measured and stored in 4% paraformaldehyde for histopathology studies or frozen at −20 °C for western blot analysis. The concentration of HSP27 in serum from control mice was measured by ELISA at 3 months after the transplantation. For bone marrow analysis, femurs were collected, fixed in 4% phosphate-buffer formalin, embedded in paraffin and sectioned. Tissue sections were stained with haematoxylin and eosin for morphology analysis, which was performed in a blinded fashion. Images were organized in folders identified by letters by one person and quantified by another. A minimum of $n$ =5–9 mice were used for each experiment, which was sufficient to provide adequate power for these experiments (inVivoStat analysis).

**Patient's samples.** Patient samples (serum, blood and histological section) were obtained from the University Hospital of Dijon and Gustave Roussy (France). The study was conducted in accordance with the Declaration of Helsinki with an approved written consent form for each patient (CPP ESTI: 2014/39 ; N°ID: 2014-A00968-39), and approval was obtained from the local ethics committee ESTI (license: NCT02873832).

**Flow cytometry.** To determine the number of megakaryocytes in the bone marrow and spleen, the cells were isolated and resuspended in lysing buffer (ACK Lysing Buffer, Bio Whittaker, Lonza) followed by incubation at RT for 10 min. After RBC lysis, cells were pelleted by centrifugation, resuspended in FACS staining medium (PBS+5% FCS) and filtered through 100 μm cell strainer (Falcon, #352360). Cells were stained with biotin-labelled mouse lineage antibodies (anti-Ter119 (clone Ter119, #116203), anti-B220 (clone RA36B2, #103203); anti-CD4 (clone GK1.5, #100403), anti-CD8 (clone 53-6.7, #100703), anti-Gr1 (clone RB6-8C5, #108403); anti-Mac1 (clone M1/70, #101203) and APC-Cy7 anti-c-kit (clone 2B8, #105825), PE-Cy7 anti-sca-1 (clone D7, #108113), APC anti-CD150 (clone TC15-12F12.2, #115909) and PE anti-CD41 (clone MWReg30, #133905) (Biolegend) for 30 min. After staining, cells were washed with FACS staining medium, pelleted down by centrifugation (1200 rpm, 5 min) and stained with streptavidin Pacific-Blue Conjugate (#1094418, Invitrogen) for 30 min followed by second wash with FACS staining medium. The cells were then pelleted down by centrifugation (1200 rpm, 5 min), resuspended in FACS staining medium containing SYTOX blue dead cell stain (S34857, Life Technologies) and acquired on LSRFortessa™ (BD Biosciences) (FACs gating strategy, Supplementary Fig. 9a).

Peripheral blood mononuclear cells (PBMCs) from peripheral blood or cord blood were collected in EDTA tubes and isolated by Ficoll-Paque density gradient centrifugation. PBMCs were immuno-stained with anti-CD34-PE-Cy7 and anti-CD45-APC antibodies in stain Buffer (BD Biosciences) for 45 min and washed. The cells were fixed and permeabilized in commercial solution (BD Biosciences) and then stained using anti-HSP FITC/Alexa 488 or anti-mouse IgG FITC control antibody for 30 min. After washing, the cells were analysed by flow cytometry. The CD45$^{low}$/CD34$^{high}$ population was gated and the median fluorescence intensities of intracellular HSPs were evaluated (FACs gating strategy, Supplementary Fig. 9b).

**Purification of CD34+ cells and erythroid progenitor assays.** Mononuclear cells from MPN patients were obtained over a Ficoll density gradient and CD34+ cells were purified by a double-positive magnetic cell sorting system (AutoMACS, Miltenyi Biotec). CD34+ cells were cultured 2 days in a liquid culture system stimulating erythropoiesis, in the presence of 50 ng ml$^{−1}$ recombinant human stem cell factor (Amgen, Thousand Oaks, CA, USA), 50 U ml$^{−1}$ interleukin-3 (Novartis, Basel, Switzerland) and 3 U ml$^{−1}$ erythropoietin (EPO; Orthobiotech, Paris, France). Cells were then transduced with lentiviral vector encoding shRNA-HSP27 or control vector (CTL). The CD34+/GFP+ were sorted 24 h later and 750–1000 cells were plated in duplicate in semi-solid conditions (methylcellulose) (H4230, Stem Cell Technologies) supplemented with interleukin-3, stem cell factor and EPO. BFU-E colonies were counted 14 days later and were plucked from methylcellulose for genotyping. The *JAK2V617F* mutational status was analysed by quantitative real-time PCR (qRT-PCR), using fluorescent competitive probes.

**Immunoassays.** Blood samples were centrifuged at 20 °C for 15 min at 2500 × $g$, and serum aliquots were stored at −80 °C for subsequent analysis. Since some HSPs are known to be unstable in plasma, we used serum for the analysis[75]. Serum levels of HSP70, HSP90α and HSP27 in patients and serum level of HSP27 in mice were quantified using specific enzyme-linked (ELISA) kits according to the manufacturer's protocol (human HSPs: ADI-EKS-715, ADI-EKS-895 ADI-EKS-500, (Enzo Life Sciences); mouse HSP27: SEA693Mu, (Cloud Clone Corp)). Before quantification, sera were diluted at 1:50 for HSP90, 1:4 for HSP70 and 1:10 dilution for HSP27 or 1:2 to detect mouse HSP27.

**Immunohistochemistry.** Immunochemistry of patient biopsies was performed using an automated Leica Bond Max on formalin-fixed, paraffin-embedded spleen sections (5 μm). Sections were deparaffinised and rehydrated through a graded series of xylene–ethanol baths. Antigen retrieval was performed using Citrate Buffer pH 6.0 for 20 min at 95 °C. Tissue sections were incubated with primary antibody for 1 h at room temperature. The antibodies were used in a 1:100 dilution for anti-HSP27 (clone G3.1, SPA-800); 1:500 dilution for anti-HSP70 (clone

C92F3A-5, ab47455); and 1:500 dilution for anti-HSP90 (clone D7a, ab59459). Visualization was performed using Novolink™ Polymer Detection Systems (Leica Biosystems). Tissue sections were counterstained with haematoxylin, dehydrated and mounted. Analysis was performed in a blinded fashion. Images were organized in folders identified by patient ID by one person and quantified by another.

**Statistics**. Data are displayed as means ± s.e.m. or +s.d. Statistically significant differences between two groups was assessed using the unpaired $t$ test with Welch's correction (for CD34$^+$ HPCs from patients), the Mann–Whitney test or Student's $t$ test depending on the distribution of the tested population. Similarity of variance was tested using GraphPad PRISM (San Diego, CA, USA) before the application of any statistical test. $P$ values less than 0.05 were considered significant.

**Data availability**. The authors declare that the other data supporting the findings of this study are available within the paper (and its supplementary information files).

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

## Acknowledgements

The authors thank the patients and healthy donors for participating and donating samples to make this research possible. The authors thank the Cellimap plateform for immunohistochemistry, Dr. Kevin Cheesman and Mr. Philip Bastable (University Hospital, Dijon, France) for revising the manuscript and Dr Vincent Ribrag for patients' serum samples. This work was funded by "La Ligue contre le cancer", the "Association pour la Recherche sur le Cancer", the association "Laurette Fugain" and the GR-Ex 2014 and the association "Tulipes contre le cancer".

## Author contributions

M.S. designed the majority of experiments, performed the research and analysed data; N.P., M.C. and G.C. performed BLI experiments, ELISA and immunoblotting; S.C. designed the PLA study; L.K., J.L.V. and C.L. generated the mouse models; F.V. analysed the mouse immunohistochemistry studies; I.P. and M.M. performed the erythroid progenitor assay. F.R.-L. gave us the domain of CBP, M.G. gave the drug (OGX-427); M.L.C., J.N.B., S.A., P.S., S.R., L.M. and I.P. collected patient samples; A.d.T., F.G. and C.G. directed the work, supervised the experiments and wrote the paper; I.P., R.S. and V.M. belong to the advisory PhD committee and revised the manuscript. All authors have seen and approved the final version of the manuscript before submission.

## Additional information

**Competing interests:** The authors declare no competing interests.

