## [Peer Review File(PDF 444 kb) · Nature Communications]

Reviewers' comments:

Reviewer #1 (Remarks to the Author):

In this paper the authors evaluate novel pathogenic mechanisms for the myeloproliferative neoplasm myelofibrosis. Their data suggests that HSP27 may have a role in this condition. They were able to observe that OGX-427 a specific inhibitor of HSP-27 reduced some disease features in a murine model of TPO induced MF. They went on to show a role for HSp-27 has a role in JAK/STAT pathway. Moving forward into patient samples increased expression of HSP-27 was shown in CD34 cells and in serum.

I enjoyed this paper and found in general it was well-written and makes an important and novel contribution to the field.

I have a few minor comments but two suggestions for further data that could strengthen the work.

The data would be further strengthened if the observations were extended to a different MPN murine model, only this TPO-driven model was evaluation.
Also have the authors considered doing further work on human cells for example the effect of HSP-27 inhibition upon EEC formation.

Figures 3b and c could be improved with more images and higher resolution

Minor comments:

Lines 69-70 reference myelofibrosis as a reactive phenomenon e.g. after infections or hodgkins disease. This is rather confused in this section with myelofibrosis as a disease (e.g. the MPN) rather than a secondary process.

Line 128 please spell out the abbreviation ICY

Line 131 purchase should read purchased.

Claire Harrison

Reviewer #2 (Remarks to the Author):

Overall, this study by Sevin et al. meets the general criteria for publication in Nature Communications. They present novel results with the first evidence for direct regulatory interactions between HSP27 and JAK2/STAT5 in a mouse model of myelofibrosis, and present consistent correlations of elevated HSP27 in patient samples. However, the main weakness of the manuscript is the lack of significant experimental details that are important for reproducing the work and for providing confidence in key results. The authors should already have most of the necessary details available. However, regarding the BLI experiments that are critical to demonstrating specific, direct interactions of HSP27 with JAK2 and STAT5A/B, an important control is missing that could be completed within a few days, as long as the protein samples are on hand. Thus, I recommend that the editors "invite the authors to revise their manuscript to address specific concerns before a final decision is reached". Specific concerns and the requested control experiment are listed below.

BLI experiments.

- Recombinant proteins were used for BLI, but no details are given for the expression or purification of these proteins: HSP27; JAK2; Stat5A; Stat5B; a domain of HAT CBP as negative control, but which domain? (See pg 4, lines 119-120). Please provide these details.
- Please specify the buffer conditions and temperature used for BLI assays.
- HSP27 was biotinylated as the ligand to immobilize on streptavidin (SA) sensors. Please provide details of the reagent and reaction conditions used for biotinylation (i.e., molar ratio of reagent to HSP27, time and temperature of reaction).
- Biotinylated HSP27 was loaded on SA sensors at 10 ug/ml, but no other details were given. A supplementary figure of 'raw' BLI data showing the loading step is preferred, but at least provide the time of the step and the average increase in BLI signal (in nm) that occurred over the loading step.
- For the BLI results, Fig. 2c and Fig. S2c, the sole negative controls were for binding of a different protein (HAT CBP domain) to SA sensors loaded with HSP27-biotin. However, not all proteins exhibit the same type or degree of non-specific binding to BLI sensors. In fact, the kinetic curves for all 3 analyte proteins show biphasic dissociation, which could be due to significant non-specific binding. Thus, distinct control assays are needed in which the highest and lowest concentration (15, 125 nM) of each analyte protein (JAK2, Stat5A, or Stat5B) are incubated with SA sensors that are NOT loaded with HSP27-biotin. Even more appropriate, the SA sensors could be loaded with free biotin or biocytin instead. With 8 parallel sensors on the Octet RED system, these controls could all be done in 2 assay runs, including a positive control of an HSP27-biotin-loaded sensor for the highest concentration of each analyte protein.

Western blot analysis: please provide the conditions used for transfer of proteins from the gel to the PVDF membrane (transfer buffer composition or ref; constant milli-Amps or voltage applied, time of transfer).

Fig. S1 – the legend (final page) has descriptions switched for panels b (survival curve) and c (blot).

Fig. S1b (survival curve) – please specify the number of mice in each group, the type of statistical analysis done, and the p-value obtained. Also, a symbol is shown (cross or 'sword') that has no explanation.

For visual evaluation of all bar-type figures by readers, it is preferable to show the error bars extending in both directions (up/down). This should be done for Figs 1b, 1c, 1d, 2a, 2b, S1a.

For Fig. 2d/e, I am not familiar with the type of data presentation shown on the right panels. The authors should include more detail on how these data points are represented and analyzed.

Reviewer #3 (Remarks to the Author):

In an original article to Nature Communications, Sevin et al. report on their findings on the role of HSP27 in myelofibrosis.

The major findings are:

- 1) Serum HSP27 levels are elevated in a TPO high model of myelofibrosis (MF)
- 2) Treatment with an antisense oligonucleotide against HSP27 (OGX-427) reduces spleen weight; causes a partial restoration of normal splenic architecture; increases red blood cells and reduces megakaryocytes in the bone marrow of TPOhigh MF mice but has no effect on peripheral blood counts; possibly improves overall survival
- 3) siRNA of HSP27 or treatment with OGX-427 reduces proliferation of HEL cells but does not

increase apoptosis

4) siRNA of HSP27 or treatment with OGX-427 decreases pSTAT5 and C-MYC levels in HEL cells but has no effect on pJAK2, JAK2 or STAT5

5) Recombinant JAK2 and STAT5A/B directly bind HSP27

6) HSP27 expression is higher in MF CD34+ cells and in MF serum as compared to healthy donors, although there is a wide range of expression and serum levels.

Overall, this is a novel and interesting study. The major caveats are that the effects of inhibiting HSP27 with OGX-427 in the TPO high MF mouse model are modest and the mechanism underlying the observed effects is unclear.

In terms of mechanism, I have the following comments:

1) In Figure 2A, the effects on cell proliferation in HEL cells are modest and the specificity of the finding is unclear. The authors should interrogate a broader range of cell lines – other JAK2V617F-mutant (e.g. SET, UKE1) and nonJAK2V617F mutant hematopoietic cells. The siRNA experiments would be improved by the use of more than one siRNA and by a rescue experiment.

2) Can the authors demonstrate physical interaction between JAK2 and/or STAT5 and HSP27 by co-immunoprecipitation?

3) The title of the manuscript is "A role for HSP27 in the regulation of JAK2 signaling in myelofibrosis" but the data presented does not show an effect on JAK2 signaling. siRNA of HSP27 or treatment with OGX-427 decreases pSTAT5 and C-MYC levels in HEL cells but has no effect on pJAK2, JAK2 or STAT5. The mechanism by which pSTAT5 levels decrease is unclear and needs to be further elucidated. The authors may also wish to consider changing the title.

Other additional comments:

1) Page 4, last sentence: "To generate the TPOhigh model, mice were lethally irradiated and animals were injected with mutated bone marrow cells as previously described" – I don't believe the donors cells are "mutated" in this assay – please edit accordingly.

2) Figure 1G - OGX-427 significantly reduces the number of megakaryocytes in the bone marrow but does not reduce fibrosis. What is the explanation for this?

3) Page 7, line 217: "JAK2-dependent leukemic HEL92.1.7 cell line" – I think HEL cells are more appropriately described as JAK2V617F-mutant rather than JAK2-dependent (at least I'm not aware of any published data indicating they are JAK2 dependent). Given the authors findings in megakaryocytes in TPO high mice and in primary MPN bone marrow samples, SET2 cells are likely a more appropriate cell line for their biochemical studies.

4) In the survival curve shown in Figure S1b, how many mice are in each group and what is the statistical measurement assessment difference in survival?

5) In Figure S2b, it would be informative to also shown HSP27 expression for each of the conditions

6) Figure 3a/b – there is a wide range of HSP27 expression and serum levels – what accounts for the observed heterogeneity e.g. stage of disease etc.

POINT-BY-POINT ANSWERS TO THE REVIEWERS

Manuscript: 17-03433

Reviewer #1

In this paper the authors evaluate novel pathogenic mechanisms for the myeloproliferative neoplasm myelofibrosis. Their data suggests that HSP27 may have a role in this condition. They were able to observe that OGX-427 a specific inhibitor of HSP-27 reduced some disease features in a murine model of TPO induced MF. They went on to show a role for HSp-27 has a role in JAK/STAT pathway. Moving forward into patient samples increased expression of HSP-27 was shown in CD34 cells and in serum.

I enjoyed this paper and found in general it was well-written and makes an important and novel contribution to the field.

I have a **few minor comments but two suggestions** for further data that could strengthen the work.

-The data would be further strengthened if the observations were extended **to a different MPN murine model**, only this TPO-driven model was evaluation.

Our answer:

As advised by the reviewer, the effect of OGX-427 was assessed on another *JAK2V617F* mouse transgenic model : *SclCre;FF1;GFP* (Kubovcakova *et al.*, Blood 2013). This model is characterized by a more pronounced MPN phenotype in all stages of the disease, including the onset and progression of myelofibrosis.

The results obtained are shown in Supplementary Fig. 2 of the revised manuscript. As for the TPO high mice model, the weight of the spleen was significantly reduced (Supplementary Fig. 2b) and the frequency of megakaryocytes progenitors showed a near-significant decrease in the spleen (Supplementary Fig. 2d). Furthermore, in this mouse model the platelet count are almost back to the normal level at the end of the treatment with OGX-427 whereas it remained very high in the control group. Altogether, these results might suggest an inhibitory effect of OGX-427 on megakaryopoiesis.

These new model and results are now described in the revised results section in page 7 lines 30; in the Supplementary material and methods section, and in the supplementary Fig. 2 legend.

We have detailed the working procedure in the supplemental data as follow:

“Bone marrow (BM) cells (2×10^6) from a wild-type control and *SclCre;FF1* mice were transplanted into lethally irradiated recipients (5 mice per group) at a 1:1 ratio as previously described (Kubovcakova *et al.*, Blood, 2013). At 6 weeks post-transplantation, a first cell blood count was performed to check and confirm MPN phenotype. Animals were then divided into 2 groups of 5 mice: one group was treated with the vehicle alone (3 times

per week) and the other mice were treated with the HSP27 inhibitor OGX-427 (10mg/kg; 3 times per week). Mice were treated during 10 weeks and cell blood count was performed every three weeks. At the end of the experiment, mice were euthanized and spleens were weighted and collected at -20°C for western-blot analysis. Mice were kept under specified pathogen-free conditions with free access to food and water. All experiments were done in strict adherence to Swiss laws for animal welfare and were approved by the Swiss Cantonal Veterinary Office of Basel-Stadt “

-Also have the authors considered doing further work **on human cells for example the effect of HSP-27 inhibition upon EEC formation.**

Our answer:

To address the reviewer's comment, we tested the effect of a shRNA-HSP27 on primary cells from patients (4 PMF and 1 PV) by performing an erythroid colony formation assay. We observed a significant reduction in BFU-E formation along with a specific effect on *JAK2V617F* cells (Supplementary Fig.3e-g). Indeed, the decrease in HSP27 expression was associated with a reduction of the heterozygotes and homozygotes *JAK2V617F* clones in 3 patients with MF and one patient with PV. It is worth to note that this effect was observed even though the expression of HSP27 was decreased by only 20%, suggesting certain specificity of HSP27 toward the *JAK2V617F* clone.

-Figures 3b and c could be improved with more images and higher resolution

Our answer:

As requested by the reviewer we have improved the resolution of the image, now shown in the revised manuscript in new Fig. 4c.

(NB: figure 3B does not exist)

Minor comments:

-Lines 69-70 reference myelofibrosis as a reactive phenomenon e.g. after infections or hodgkins disease. **This is rather confused in this section** with myelofibrosis as a disease (e.g. the MPN) rather than a secondary process.

Our answer:

As recommended by the reviewer, we have shortened the section page 3 line 5 « MF develops as a non-specific response to a variety of malignant and benign diseases especially myeloproliferative neoplasms (MPN) but can also be induced following treatment with hematopoietic growth factors like thrombopoietin (TPO) » as follows:

“MF are mostly related to myeloproliferative neoplasms (MPN) but can also be induced following treatment with hematopoietic growth factors like thrombopoietin (TPO)”

-Line 128 please spell out the abbreviation ICY

Our answer:

ICY is a software of image analysis (<http://icy.bioimageanalysis.org/>). As far as we know there is

no abbreviation available.

-Line 131 purchase should read purchased.

Our answer:

The mistake has been corrected.

Reviewer #2

Overall, this study by Sevin et al. meets the general criteria for publication in Nature Communications. They present novel results with the first evidence for direct regulatory interactions between HSP27 and JAK2/STAT5 in a mouse model of myelofibrosis, and present consistent correlations of elevated HSP27 in patient samples. However, the main weakness of the manuscript is **the lack of significant experimental details that are important for reproducing the work** and for providing confidence in key results. The authors should already have most of the necessary details available. However, regarding the BLI experiments that are critical to demonstrating specific, direct interactions of HSP27 with JAK2 and STAT5A/B, an important control is missing that could be completed within a few days, as long as the protein samples are on hand. Thus, I recommend that the editors “invite the authors to revise their manuscript to address specific concerns before a final decision is reached”. Specific concerns and the requested control experiment are listed below.

BLI experiments.

- **Recombinant proteins were used for BLI, but no details are given for the expression or purification of these proteins: HSP27; JAK2; Stat5A; Stat5B; a domain of HAT CBP as negative control, but which domain? (See pg 4, lines 119-120). Please provide these details.**

- **Please specify the buffer conditions and temperature used for BLI assays.**

- **HSP27 was biotinylated as the ligand to immobilize on streptavidin (SA) sensors. Please provide details of the reagent and reaction conditions used for biotinylation (i.e., molar ratio of reagent to HSP27, time and temperature of reaction).**

- **Biotinylated HSP27 was loaded on SA sensors at 10 ug/ml, but no other details were given. A supplementary figure of ‘raw’ BLI data showing the loading step is preferred, but at least provide the time of the step and the average increase in BLI signal (in nm) that occurred over the loading step.**

Our answer:

As requested, we have added the following technical details in the material and method section so the experiments can be easily reproduced. Biotinylated recombinant HSP27 (Enzo-Life Sciences ADI-ESP-715, low endotoxin) was used as a ligand and immobilized at 10 µg/ml on

streptavidin biosensors after dilution in PBS. Raw data for the loading step is now included in supplementary Fig. 4a-c (time of loading: 600s and average increase in BLI signal: 1nm). Interaction with desalted analytes diluted in PBS at 125nM was analysed after association (time: 600s) and dissociation (time: 600s) steps at 26°C.

Analytes used: recombinant STAT5a/b (Origen TP305753 and TP309429, respectively), JAK2 (Origen TP320503) or as a negative control the RING domain of CBP (gift from Dr. Rodriguez-Lima. Duval et *al.*, 2015) and as a positive control the alpha B crystallin (Fu, L. & Liang, J. J., 2002).

This information is now included in the revised manuscript (page 4, line 25).

Supplemental informations for the reviewer:

- **Recombinant HSP 27: (Enzo-Life Sciences ADI-ESP-715 (low endotoxin))**

Cm = 1.7 mg/ml soit CM = 62962 nM

MM : 27 000g/mol

Tampon : PBS (low endotoxin)

Source : Produced in E. coli.

- **Recombinant JAK 2 : (Origene TP320503)**

Cm = 0.1 mg/ml soit CM = 62 962 nM

MM : 130 500 g/mol

Tampon : 25 mM Tris.HCl, pH 7.3, 100 mM glycine, 10% glycerol.

Source : Recombinant protein was produced with TrueORF clone, RC220503, encoding the full-length human JAK2 with C-terminal DDK tag, from human HEK293 cells.

- **Recombinant STAT 5A: (Origene TP305753)**

Cm = 0.414 mg/ml soit CM = 4 567 nM

MM : 90 647g/mol

Tampon : 25 mM Tris.HCl, pH 7.3, 100 mM glycine, 10% glycerol.

Source: Recombinant protein was produced with TrueORF clone, RC205753, encoding the full-length human STAT5A with C-terminal DDK tag, from human HEK293 cells.

- **Recombinant STAT 5B : (Origene TP309429)**

Cm = 0.298 mg/ml soit CM = 3 316 nM

MM : 89 866 g/mol

Tampon : 25 mM Tris.HCl, pH 7.3, 100 mM glycine, 10% glycerol.

Source: Recombinant protein was produced with TrueORF clone, RC209429, encoding the full-length human STAT5B with C-terminal DDK tag, from human HEK293 cells.

- **Recombinant RING : (gift from Dr. F.Rodriguez-Lima)**

Cm = 0.91 mg/ml

MM : 13 000g/mol

Buffer: Tris HCl pH 7.5, 150 mM NaCl

Source: recombinant protein as described in Duval et al., 2015

- **In addition, as requested, the raw data for the loading step is now included in new supplementary Figure 4a-c,**

The association and dissociation steps were done as follows:

- Baseline : PBS - 60s
- Loading : : HSP 27 – 600s (the loading was stopped at 1nM)
- Baseline : - 120s
- Baseline : - 120s
- Association : - 600s
- Dissociation : - 600s
- Temperature : 26°C

- **For the BLI results, Fig. 2c and Fig. S2c,** the sole negative controls were for binding of a different protein (HAT CBP domain) to SA sensors loaded with HSP27-biotin. However, not all proteins exhibit the same type or degree of non-specific binding to BLI sensors. In fact, the kinetic curves for all 3 analyte proteins **show biphasic dissociation**, which could be due to significant non-specific binding.

Thus, distinct control assays are needed in which the highest and lowest concentration (15, 125 nM) of each analyte protein (JAK2, Stat5A, or Stat5B) are incubated with **SA sensors that are NOT loaded with HSP27-biotin.**

Even more appropriate, the SA sensors could be loaded **with free biotin or biocytin instead.** With 8 parallel sensors on the Octet RED system, these controls could all be done in 2 assay runs, including a **positive control of an HSP27-biotin-loaded sensor for the highest concentration of each analyte protein.**

Our answer:

To properly address the reviewer's comment concerning the specificity of our BLI experiments, we asked advice to a technical expert of ForteBio (BLI France). He recommended us that instead of using the unloaded tip as negative control (if it is unloaded almost everything can stick to it), to use the SA sensors loaded with an irrelevant protein like the biotinylated RING domain of CBP (gift from Dr. Rodriguez-Lima. Duval et al., 2015), which has the advantage (compared to, for instance, biocytin or free biotin) of having a molecular weight close to that of HSP27. The HSP27-loaded sensor as well as the RING-CBP loaded sensor (negative control) were then put in contact with the highest concentration of each analyte (JAK2, STAT5).

As shown in Figure 2 for reviewers, all analytes interacted with the unloaded tips (as expected by the BLI ForteBio expert). However, JAK2 and STAT5 only interacted with the tip loaded with HSP27 but not with the tip loaded with RING-CBP.

Therefore, our results demonstrate a specificity in the interaction as JAK2 and STAT5 associate to the sensor loaded with HSP27-biotin but not to the one loaded with the

irrelevant protein.

To further complete these results, we tested another negative control (HAT domain of CBP) and, as a positive control, alphaB-crystallin, known to interact with HSP27 (Fu, L. & Liang, J. J., 2002) (see Fig 2 a-c for reviewers). The results obtained show that alphaB-crystallin interacts with HSP27 within the same range than STAT5 and JAK2, while the negative control does not. These results are shown in new Supplementary Fig 4a-c of the revised manuscript and are described in page 8 line 31.

To reinforce our results, we evaluated the interaction between HSP27 and STAT5 by a third method i.e co-immunoprecipitation of both endogenous proteins (new Fig. 3b) and exogenous tagged proteins after transfection (new Supplementary Fig. 4d,e).

For the endogenous proteins, HEL92.1.7 cells were deprived of serum for 16h and then stimulated 10 min with serum or increasing doses of erythropoietin (EPO, 15 and 50 UI/ml). Then, endogenous HSP27 was immunoprecipitated from cell lysates using a mouse anti-HSP27 antibody (ADI-SPA800, Enzo Life) followed by immunodetection of endogenous JAK2 (rabbit anti-JAK2 antibody CST 3230) and STAT5 (rabbit anti-STAT5 antibody CST 9363). As shown Figure 3b, and in line with our previous results, while HSP27 interacts mainly with JAK2 in unstimulated cells, the interaction with STAT5 is more pronounced in cells stimulated with EPO (50 UI/ml).

- Western blot analysis: please provide the conditions used for transfer of proteins from the gel to the PVDF membrane (**transfer buffer composition or ref; constant milli-Amps or voltage applied, time of transfer**).

Our answer:

For the western-blot, the transfer step was performed in borate buffer (50mM TrisHCl, 50mM borate) for 1h45 at constant voltage (48V). This is now clearly stated in the material and method section of the revised manuscript (Page 4, line 18).

-Fig. S1 – the legend (final page) has descriptions **switched** for panels b (survival curve) and c (blot).

Our answer:

We apologise for this mistake that has now been corrected in the revised paper.

-Fig. S1b (survival curve) – please specify the **number of mice** in each group, the type of **statistical analysis** done, and the **p-value** obtained. Also, a **symbol is shown** (cross or 'sword') that has no explanation.

Our answer:

As advised, supplementary Fig 1b includes now the following information:

“ A Kaplan-Meier survival curve was done for the control (CTL, n=9) and treated group (OGX-427, n=9). Outcomes of the two treatments were compared using a log-Rank statistical analysis

($p=0,32$).”

The cross symbolizes the euthanasia of the animals.

It is worth noting that only a few mice died by the end of the experiment, thereby it is not possible with this number of mice to get a significant p value.

-For visual evaluation of all **bar-type figures** by readers, it is preferable to show the error bars extending in **both directions (up/down)**. This should be done for Figs 1b, 1c, 1d, 2a, 2b, S1a.

Our answer:

We have changed the error bars (extended in both directions) in all graphs from Fig 1, 2 and Supplementary Fig. 1, as advised.

-**For Fig. 2d/e**, I am not familiar with the type of data presentation shown on the right panels. The authors should **include more detail on how these data points** are represented and analyzed.

Our answer:

As requested by the reviewer, we have added more details in the legend of the new Fig 3c concerning the PLA analysis:

“(c) Immunofluorescence analysis of the endogenous interaction (red foci) of JAK2 (upper panel) or STAT5 (middle panel) with HSP27 visualized *in situ* by PLA in HEL92.1.7 cells transfected or not with a HSP27 siRNA. Nuclei were stained with DAPI (4,6 diamidino-2-phenylindole). Images were obtained using an Axio Imager 2 (Zeiss, France) at X40 magnification. Images were taken randomly and analyzed using ICY software. Cells were segmented manually, and the numbers of interaction foci in each cell were determined using the spot detector plugin. Statistical analysis (Mann and Whitney test) was carried out using GraphPad. Right panel, graphs represent quantification of the interaction of endogenous JAK2 or STAT5 with HSP27 visualized *in situ* by PLA. Each data point corresponds to an analyzed cell, placed according to the number of detected interaction foci. (lower panel) Immunofluorescence analysis of the endogenous interaction (red foci) of JAK2 with STAT5 visualized *in situ* by PLA in HEL92.1.7 cells transfected with a scramble (CTL) or a HSP27 siRNA (siHSP27). Nuclei were stained with DAPI (4,6 diamidino-2-phenylindole). Images were obtained using an Axio Imager 2 (Zeiss, France) at X63 magnification and analysed using ICY software as in upper panel. Right panel, graphs represent quantification of the interaction of endogenous JAK2 with STAT5 visualized *in situ* by PLA.” ☐

Reviewer #3:

In an original article to Nature Communications, Sevin et al. report on their findings on the role of HSP27 in myelofibrosis.

The major findings are:

- 1) Serum HSP27 levels are elevated in a TPO high model of myelofibrosis (MF)
- 2) Treatment with an antisense oligonucleotide against HSP27 (OGX-427) reduces spleen weight; causes a partial restoration of normal splenic architecture; increases red blood cells and reduces megakaryocytes in the bone marrow of TPOhigh MF mice but has no effect on peripheral blood counts; possibly improves overall survival
- 3) siRNA of HSP27 or treatment with OGX-427 reduces proliferation of HEL cells but does not increase apoptosis
- 4) siRNA of HSP27 or treatment with OGX-427 decreases pSTAT5 and C-MYC levels in HEL cells but has no effect on pJAK2, JAK2 or STAT5
- 5) Recombinant JAK2 and STAT5A/B directly bind HSP27
- 6) HSP27 expression is higher in MF CD34+ cells and in MF serum as compared to healthy donors, although there is a wide range of expression and serum levels.

Overall, this is anovel and interesting study. The major caveats are that the effects of inhibiting HSP27 with OGX-427 in the TPO high MF mouse model are modest and the mechanism underlying the observed effects is unclear.

In terms of mechanism, I have the following comments:

1) In Figure 2A, the effects on cell **proliferation in HEL cells are modest** and the specificity of the finding is unclear. The authors should interrogate a **broader range of cell lines** – other JAK2V617F-mutant (e.g. SET, UKE1) **and nonJAK2V617F** mutant hematopoietic cells. The siRNA experiments would be improved by the use of **more than one siRNA and by a rescue** experiment.

Our answer:

As requested, to confirm the effect of HSP27 on the cells' proliferation, the revised manuscript now includes another siRNA against HSP27 and two additional cell lines: SET2 (*JAK2V617F* positive) and K562 (*JAK2V617F* negative). As shown in new Fig 2a, while depletion of HSP27 has no effect on K562 proliferation, it impairs the proliferation of SET2 cells and the effect is stronger than that observed for the HEL92.1.7 cells. These results obtained with two different siRNAs are in accord with those obtained with the OGX-427 that, like the siRNA, affects the translation of HSP27.

A similar inhibition in cell proliferation was also obtained when HSP27 was knock down with a shRNA in HEL92.1.7 cells (new Supplementary Fig. 3b). This inhibition of cell proliferation observed in the shRNA HSP27 infected cells could be partially rescued by transient transfection with a HSP27 plasmid (Fig. 3 for the reviewer).

Altogether, we show using three different approaches (i.e OGX-427, siRNA, shRNA) that HSP27 play a role on the proliferation of MPN cells, thus reinforcing the relevance of our results.

2) Can the authors demonstrate **physical interaction between JAK2 and/or STAT5** and HSP27 by co-immunoprecipitation?

Our answer:

As requested, we have now determined the interaction between HSP27 and STAT5 by co-immunoprecipitation of both the endogenous proteins (new Fig 3b) and the exogenous tagged proteins after transfection (new supplementary Fig 4d-e). For the association between the endogenous proteins, HEL92.1.7 cells were first deprived of serum for 16h and then stimulated 10 min with serum or increasing doses of erythropoietin (EPO, 15 and 50 UI/ml) to stimulate the JAK2/STAT5 signalling. Then, endogenous HSP27 was immunoprecipitated from cell lysates using a mouse anti-HSP27 antibody followed by immunodetection of endogenous JAK2 and STAT5. As shown Fig 3b, and in line with our previous results, while HSP27 interacts mainly with JAK2 in unstimulated cells, HSP27 interaction with STAT5 was more pronounced in cells stimulated with EPO (50 UI/ml).

Although in a more artificial context, an interaction between HSP27 and JAK2 and STAT5 was also observed with the exogenous (tagged) proteins (new Supplementary Fig. 4d-e).

3) The title of the manuscript is “A role for HSP27 in the regulation of JAK2 signaling in myelofibrosis” but the **data presented does not show an effect on JAK2 signaling**. siRNA of HSP27 or treatment with OGX-427 decreases pSTAT5 and C-MYC levels in HEL cells but has no effect on pJAK2, JAK2 or STAT5. The mechanism by which **pSTAT5 levels decrease is unclear and needs to be further elucidated**. The authors may also wish to consider changing the title.

Our answer:

To avoid any misunderstanding, we have changed the title as follows: “Impact of HSP27 in myelofibrosis: a new partner of JAK2/STAT5 ”

Our results show a direct and specific interaction between HSP27 and JAK2/STAT5 by different approaches *in vitro*, *in cellulo* and *in situ* (BLI, co-IP and PLA, respectively). We also show that when HSP27 is depleted, the amount of JAK2/STAT5 complexes decreases (new Fig 3c, lower panel). Since HSP27 is a well-known chaperon, our results suggest that HSP27 could act as a scaffold protein or favour the molecular assembly JAK2/STAT5 (as it has been described for other HSP27 client proteins (See reference 32-34 from manuscript).

Whether HSP27 has another effect in JAK2/STAT5 signalling, for instance in the expression of STAT5 (STAT5 expression in the megakaryocytic cell line SET2 is slightly diminished by OGX-427) will be explored in depth in another work.

To further test the impact of HSP27 on JAK2, we studied the effect of HSP27 depletion on erythroid colony formation. Using a shRNA lentivirus to knock-down HSP27, we observed a significant reduction in BFU-E formation along with a specific effect on *JAK2V617F* cells in patients (4 PMF and 1 PV) (Supplementary 3e-g).

Other additional comments:

1) **Page 4, last sentence:** “To generate the TPO^{high} model, mice were lethally irradiated and animals were injected with **mutated bone marrow cells** as previously described” – I don’t believe the donors cells are “mutated” in this assay – please edit accordingly.

Our answer:

The sentence Page 5 line 22 “To generate the TPO^{high} model, mice were lethally irradiated and animals were injected with mutated bone marrow cells as previously described” has been substituted by " To generate the TPO^{high} model, mice were lethally irradiated and animals were injected with transduced bone marrow cells (retrovirus MPZenTPO)" as previously described^{22,23}.

2) **Figure 1G** - OGX-427 significantly reduces the number of megakaryocytes in the bone marrow but **does not reduce fibrosis. What is the explanation for this?**

Our answer:

Justification for this is questionable. The process of fibrosis in the TPO high model is very rapid and irreversible. Therefore, it might be that the MF is already initiated at the time of the treatment, which begins one month after transplantation (one month is the minimum time needed for the mice to recover before any treatment). Another explanation is that since the depletion of HSP27 induced by OGX-427 is only partial, it is possible that a more pronounced decrease in HSP27 is necessary to significantly impair the fibrotic process.

3) **Page 7, line 217:** “JAK2-dependent leukemic HEL92.1.7 cell line” – I think HEL cells are more appropriately described as JAK2V617F-mutant rather than JAK2-dependent (at least I’m not aware of any published data indicating they are JAK2 dependent). Given the authors **findings in megakaryocytes in TPO high mice and in primary MPN bone marrow samples, SET2 cells are likely a more appropriate cell line for their biochemical studies.**

Our answer:

As advised, the sentence in page 7 line 30 “JAK2-dependent leukemic HEL92.1.7 cell line” has been substituted by “JAK2V617F-positive HEL92.1.7 cells”

As requested, to evaluate the effect of HSP27 our studies include now another siRNA^{HSP27} and additional cell lines: SET2 (*JAK2V617F* positive) and K562 (*JAK2V617F* negative). As shown in new Fig 2b of the revised manuscript, while depletion of HSP27 has no effect on K562 proliferation, it impairs the proliferation of SET2 cells and this effect is stronger (as the reviewer foresaw) than that observed in HEL92.1.7 cells.

4) In the **survival curve** shown in **Figure S1b**, how **many mice** are in each group and what is the **statistical** measurement assessment difference in survival?

Our answer:

We now provide the following information in Supplementary Fig 1b:

“A Kaplan-Meier survival curve was performed for the control (CTL, n=9) and treated group (OGX-427, n=9) Outcomes of the two treatments were compared using a log-Rank statistic analysis (p=0,32).”

It is worth noting that only a few mice died by the end of the experiment, thereby it is not possible with this small number of mice to get a significant p value.

5) In **Figure S2b**, it would be informative to also show **HSP27 expression** for each of the conditions

Our answer:

Expression of HSP27 for each of the conditions is now included in supplementary Fig 3c.

6) **Figure 3a/b** – there is a wide range of HSP27 expression and serum levels – **what accounts for the observed heterogeneity e.g. stage of disease etc.**

Our answer:

This is a question that deserves to be studied in depth. MF patients are heterogeneous in terms of (i) the type of driver (*JAK2*, *CALR* and *MPL*) or secondary mutations, (ii) the nature of MF, primary or secondary to MPN, (iii) the stage of the disease depending on the International Prognostic Scoring System (Spivak, 2017*).

So far, in our study, we didn't see a correlation between the type of driver mutation and HSP27 levels.

However, when the patients were separated into two groups according to the primary or secondary feature of the myelofibrosis (in other words, primary MF and MF post MPN) HSP27 serum levels within patients of the same group were more homogenous, HSP27 levels being higher in the primary fibrosis group (Fig 4b, for reviewer). This difference in HSP27 level between the patients' groups was more striking if HSP27 levels were analysed in circulating CD34+ cells (Fig. 4a for reviewers).

*Spivak JL. *Myeloproliferative Neoplasms. N Engl J Med. 2017 Jun 1;376(22):2168-2181*

Reviewers' comments:

Reviewer #1 (Remarks to the Author):

This paper is much improved, the results clarified and indeed extended.

Reviewer #2 (Remarks to the Author):

Sevin et al. have done an admirable job in responding to the concerns and requests of reviewers. In particular, results from the added experiments with other murine MPN cell lines and a second siRNA for HSP27 (Fig. 2 and supplement) strengthen their basic conclusions that HSP27 impacts the JAK2/STAT5 pathway. New experiments with shRNA against HSP27 (Supp. Fig. 3) are also supportive, and I suggest that the rescue experiment (Fig. 2 for the reviewer) should be included in the supplementary Figs. For the BLI experiments (Fig. 3), the additional results shown in Suppl. Fig. 4 and in "Figure 1 for the reviewer" are sufficient to confirm direct, specific interactions of JAK2 and STAT5 with immobilized HSP27. Added co-IP assays with cell lysates (Fig. 3b) also support interactions of HSP27 with JAK2 and STAT5, although the authors do not comment on the potentially interesting result that the JAK2/HSP27 interaction is more prominent in the control than after EPO treatment. I believe this revised study has sufficient rigor and novel impact that it should be accepted for publication, with the minor revisions noted below.

-Lines 111-119, details are needed for the incubation of primary antibodies with the blotted membranes.

-Lines 285-289, p-values need to be provided here or in the Fig. 3 legend.

-Line 514, should note NT abbreviation: "transfected or not (NT)..."

-For Fig. 3a, the legend should state that the control sample uses 125 nM CBP-RING domain.

-The important BLI technical controls shown in "Figure 1c for the reviewer", with immobilized RING instead of HSP27, should be included, preferably in Suppl. Fig. 4, or at least stated clearly in the methods (lines 121-131).

-For Fig. 3b, the legend should clarify the conditions of the sample lanes labeled CTL and +SVF.

-Line 282 should note that some results in Suppl. Fig. 4d are from in vitro expression (RLL) rather than from transfected cells.

Reviewer #3 (Remarks to the Author):

Following up the points are raised in my initial review:

Point 1:

Authors tested SET2 and K562 cells. Effects on proliferation in SET3 cells are modest, just as in HEL cells.

The authors refer to a rescue experiment in Figure 3 but I do not see this?

Point 2:

Figure 3 Co-IP – poorly quality blot (esp for STAT5), I'm not convinced of the reported findings.

Point 3: The mechanism by which pSTAT5 levels decrease is unclear and needs to be further elucidated.

The authors do not address this question in their response.

Other comments:

The author's data indicate there is no survival advantage for treatment and this is as a result of low mice numbers. If the authors are under powered to assess for a difference in survival, the solution is to treat more mice!

POINT-BY-POINT ANSWERS TO THE REVIEWERS

We thank the reviewers for all their comments that have allowed us, we believe, to improve the manuscript. In addition to all the corrections requested, in this revised manuscript we have been able to fully complete our second murine model of myelofibrosis (*JAK2V617F*), more adapted for medullar fibrosis studies. Our results obtained concerning the medullar fibrosis are so conclusive, that we have included in the revised manuscript the results obtained with this second model as a main Figure of the paper (new Fig. 2).

Reviewer #1 (Remarks to the Author):

This paper is much improved, the results clarified and indeed extended.

Reviewer #2 (Remarks to the Author):

Sevin et al. have done an admirable job in responding to the concerns and requests of reviewers. In particular, results from the added experiments with other murine MPN cell lines and a second siRNA for HSP27 (Fig. 2 and supplement) strengthen their basic conclusions that HSP27 impacts the JAK2/STAT5 pathway. New experiments with shRNA against HSP27 (Supp. Fig. 3) are also supportive, and I suggest that the rescue experiment (Fig. 2 for the reviewer) should be included in the supplementary Figs. For the BLI experiments (Fig. 3), the additional results shown in Suppl. Fig. 4 and in Figure 1 for the reviewer are sufficient to confirm direct, specific interactions of JAK2 and STAT5 with immobilized HSP27. Added co-IP assays with cell lysates (Fig. 3b) also support interactions of HSP27 with JAK2 and STAT5, although the authors do not comment on the potentially interesting result that the JAK2/HSP27 interaction is more prominent in the control than after EPO treatment. I believe this revised study has sufficient rigor and novel impact that it should be accepted for publication, with the minor revisions noted below.

- I suggest that the rescue experiment (Fig. 2 for the reviewer) should be included in the supplementary Figs.

- Our answer:

As requested by the reviewer, the rescue experiment is now shown in Supplementary Fig. 2d

- the authors do not comment on the potentially interesting result that the JAK2/HSP27 interaction is more prominent in the control than after EPO treatment

- Our answer:

We thank the reviewer for this interesting remark. The more prominent JAK2/HSP27 interaction upon EPO exposure might be explained by the fact that EPO strongly increases the phosphorylation of HSP27 (please, see Figure below), which will impact on its conformation and thereby on its interaction with its partners. How the phosphorylation of HSP27 affects

myelofibrosis and the therapeutic interest of regulating HSP27 phosphorylation in this pathology will be the subject of our next work.

HSP27 and STAT5 phosphorylation levels increase upon EPO treatment. HEL 92.1.7 were deprived over-night and then treated with EPO for 30 min (50UI/ml). Cells lysates, from the co-IP Fig.5b, were subjected to migration 8-12% and blotted using STAT5, Phospho-STAT5, HSP27 and Phosphoserine-15 HSP27 antibodies. Actin serves as a loading control.

-Lines 111-119, details are needed for the incubation of primary antibodies with the blotted membranes.

Our answer:

For the western-blot, the incubation of primary antibodies with the blotted membrane was done at 4°C overnight. This is now clearly stated in the material and method section of the revised manuscript (page 4-5, lines 127-129).

-Lines 285-289, p-values need to be provided here or in the Fig. 3 legend.

Our answer:

The p value (**** $p \leq .0001$) has been added in the new Fig. 5 legend.

-Line 514, should note NT abbreviation: “transfected or not (NT)...”

Our answer:

We have now defined the abbreviation NT (page 22 line 662).

-For Fig. 3a, the legend should state that the control sample uses 125 nM CBP-RING domain.

Our answer:

In the legend of new Fig 5a (old Fig 3a), the characteristics of the control we used for the experiment are now clearly indicated (i.e CBP-RING domain-125 nM).

-The important BLI technical controls shown in “Figure 1c for the reviewer”, with immobilized RING instead of HSP27, should be included, preferably in Suppl. Fig. 4, or at least stated clearly in the methods (lines 121-131).

Our answer:

As requested by the reviewer, we included in the Supplementary Fig. 3d-e the BLI technical

controls.

-For Fig. 3b, the legend should clarify the conditions of the sample lanes labeled CTL and +SVF.

Our answer:

The legend of new Fig. 5b (old Fig. 3b) now clearly indicates the conditions of the sample lanes (i.e. CTL sample corresponds to the unstimulated cells starved for 16h and the +SVF sample corresponds to the cells starved 16h and then stimulated with SVF (10%) for 30 min).

-Line 282 should note that some results in Suppl. Fig. 4d are from in vitro expression (RLL) rather than from transfected cells.

Our answer:

We apologise for this mistake that has now been corrected in the revised paper in new Fig. 4b.

Reviewer #3 (Remarks to the Author):

Following up the points are raised in my initial review:

Point 1:

Authors tested SET2 and K562 cells. Effects on proliferation in SET2 cells are modest, just as in HEL cells.

Our answer:

We observed a reduction in HEL92.1.7 cells proliferation of about 20%, whereas for the SET2 cells this reduction was of about 50% (almost the double) (Fig. 3a). Certainly, the effect is not complete, but it has been consolidated by our results in progenitor cells from patients with myelofibrosis. Indeed, we also show a significant effect of an shRNA of HSP27 on the number of primary erythroid progenitors (BFU-E) from myelofibrosis patients by colony assay (about 40% of reduction, Supplementary Fig 2f-h). Importantly, this is associated with a specific effect on JAK2V617F cells in patients (4 PMF and 1 PV).

Altogether these results suggest that the observed effect of HSP27 depletion on the proliferation of the different cell lines might have a relevant clinical impact.

The authors refer to a rescue experiment in Figure 3 but I do not see this?

Our answer:

The rescue experiment, which was included in our previous revised paper in the Figures for the reviewers, is now shown in Supplementary Fig. 2d.

- Rescue experiment:

HEL92.1.7 cells depleted in HSP27 after 72h infection with a shRNA HSP27, were then transfected with a HSP27 plasmid. The proliferation of the cells was determined 24h after cell transfection using a XTT assay kit.

Our result shows that even with just a two-fold increase in HSP27, we are able to improve the

proliferation of HEL92.1.7 cells (Supplementary Fig. 2d). Altogether, this experiment consolidates our results shown in Figure 3a that are in agreement with the proliferative role of HSP27 already reported in the literature (Zhang et al., 2015).

Point 2:

Figure 3 Co-IP; poorly quality blot (especially for STAT5), I'm not convinced of the reported findings.

Our answer:

As advised by the reviewer, we repeated the co-immunoprecipitation experiment with the endogenous proteins. Although modest, we still detect an interaction not only in HEL92.1.7 but also in SET2 cell lines (new Fig. 5c and Supplementary Fig. 4a). This result, often seen for endogenous co-IP, may be explained by the low amount of the endogenous proteins together with the dynamic of the reaction. It should be mentioned that the interaction of HSP27 with JAK2 and STAT5 is confirmed in this manuscript by co-IP with exogenously expressed tagged proteins (Supplementary Fig. 4b,c), *in vitro* by BLI (Fig. 5a) and *in situ* within the cells by PLA (Fig 5d).

Point 3:

The mechanism by which pSTAT5 levels decrease is unclear and needs to be further elucidated. The authors do not address this question in their response.

Our answer:

We apologize we did not properly address this question. To shed some light on the mechanism by which HSP27 impacts phosphorylated STAT5 level, we have now performed both STAT5 phosphorylation and de-phosphorylation assays. First, we found in our *in vitro* phosphorylation assays with recombinant proteins that HSP27 did not affect the phosphorylation of STAT5 by JAK2 (new Fig. 4b).

Based on this, we sought to determine whether HSP27 may be involved in the regulation of phosphatases-induced STAT5 de-phosphorylation, as it has been previously described for the protein SRSF10 (formerly SRp38) (Shi et al., 2011). The phosphorylation of STAT5 has been shown to be tightly regulated by phosphatases, the most important one being the cytoplasmic tyrosine phosphatase, SHP2 (Chen et al., 2003, 2004, Yu et al., 2013). Therefore, we performed *in vitro* assays in which we first induced the phosphorylation of STAT5 by the recombinant JAK2 in presence of ATP and then, we added either the recombinant phosphatase SHP2 alone or together with HSP27. The reaction was stopped and the level of phosphorylated STAT5 was assessed by western blot. Our results show that the de-phosphorylation of STAT5 is indeed significantly impaired in presence of HSP27 (new Fig. 4a).

To confirm the protective role of HSP27 toward STAT5 de-phosphorylation, we compared the kinetics of de-phosphorylation of STAT5 in HEL92.1.7 cells, depleted or not for HSP27, following

JAK2 inhibition using the AG-490 inhibitor (Gautier et al., 2012). As shown in new Fig. 4c, AG-490 induced a faster de-phosphorylation of STAT5 in HEL92.1.7 cells depleted for HSP27 than in control cells.

Altogether, using *in vitro* and *in cellulo* approaches, our results unveil a protective role of HSP27 on the de-phosphorylation of STAT5 induced by the tyrosine phosphatase SHP2.

Other comments:

The author data indicate there is no survival advantage for treatment and this is as a result of low mice numbers. If the authors are under powered to assess for a difference in survival, the solution is to treat more mice!

Our answer:

Concerning the comment of the reviewer about increasing the number of animals used in order to study the animals' survival (in each experiment we used 10 animals per group), we are confronted to the ethically accepted protocols used in this study. Indeed, to study OGX427 effect in myelofibrosis development, we euthanized the animals about 5 months after the beginning of the disease. To wait until the disease provokes the death of the animal in these models of myelofibrosis can be long and, most important, very painful for the animals at the late stages of the disease. The protocol accepted by our animal ethics committees was set to study the development of the disease and not the animals' survival. That is why we had very few animals' death (in total n=4 for the TPO^{high} model and n=0 for the JAK2V617F murine model) before the planned euthanasia, which allowed us to analyse in the different organs the evolution of the disease. An increase in the number of animals will not change the fact that the animal protocols of this study are not adequate to study the animals' death provoked by the disease.

This is now better explained in the revised paper (please, see page 9, lines 293-298, of the revised manuscript).

Reviewers' Comments:

Reviewer #2:

Remarks to the Author:

Sevin et al. have provided satisfactory answers to all of this reviewer's previous questions. I believe that they have also adequately answered the questions of the other reviewers. Most of their new results contribute effectively to strengthen and extend their conclusions. I have one issue below with details of the new Fig. 4b, although it is not critical to the authors' main conclusions. I also have three minor recommendations for clarity, as listed below. Thus I highly recommend publishing this study with these minor revisions.

Main point about results.

1. In Fig. 4b., the detection of total STAT5 appears weaker and shows more diffuse bands compared to the analogous panel of Fig. 4a. This would be challenging for quantifying the STAT5 signals, and could contribute to the large error bars on the bar chart for the ratios of Ph-STAT5/STAT5. However, detection of the Ph-STAT5 band appears 'cleaner'. Presuming that the same lot of STAT5 (produced by in vitro transcription/translation) was used for all 4 samples in a specific experiment, the total amount of STAT5 should be the same for all 4 samples. Thus, if all 4 samples were detected on the same Ph-STAT5 blot, the intensities of the Ph-STAT5 bands should be directly comparable. If so, I suggest that the authors include the ratio of signals for [Ph-STAT5+SHP2]/[Ph-STAT5-] for the samples with and without HSP27. This ratio might show a more reproducible effect of HSP27 on dephosphorylation.

Other minor issues of clarity

2. Regarding Fig. 3, I did not realize before that the 2 actin rows probably indicate controls for 2 distinct blots. If so, it appears that the top 2 rows of samples (above the 1st actin row) are from 1 experiment and the 4 rows of samples above the bottom actin row are from a 2nd experiment. If this is correct, the authors could make this clearer by adding a dotted line below the 1st actin row, or by adding brackets to 1 side to show the grouping of samples from each experiment.

3. Fig. 4 legend (pg 21, lines 640-641). The text is somewhat confusing. I presume the bars represent a normalized ratio of Ph-STAT5/STAT5 from the blots above. However, since they also show blots detecting SHP2rec, JAK2rec, and HSP27, the description on line 641 should be more explicit. Suggested text: "Bar graphs show normalized ratios of Ph-STAT5/STAT5 bands quantified from the western blots (n=3 ..."

4. Supplementary Fig. 4b and legend. Labels for the sample lanes in Supplementary Fig. 4b include the designations C2 and C1 (2nd and 3rd lanes, respectively). I presume these indicate 2 distinct sample concentrations loaded, but this should be noted in the figure legend.

Reviewer #3:

Remarks to the Author:

The authors have adequately addressed all of my questions.

POINT-BY-POINT ANSWERS TO THE REVIEWERS

We thank the two reviewers for all their useful comments that have allowed us to improve the manuscript. We have made all the corrections requested.

Reviewer #2 (Remarks to the Author):

Sevin et al. have provided satisfactory answers to all of this reviewer's previous questions. I believe that they have also adequately answered the questions of the other reviewers. Most of their new results contribute effectively to strengthen and extend their conclusions. I have one issue below with details of the new Fig. 4b, although it is not critical to the authors' main conclusions. I also have three minor recommendations for clarity, as listed below. Thus I highly recommend publishing this study with these minor revisions.

Main point about results.

1. In Fig. 4b., the detection of total STAT5 appears weaker and shows more diffuse bands compared to the analogous panel of Fig. 4a. This would be challenging for quantifying the STAT5 signals, and could contribute to the large error bars on the bar chart for the ratios of Ph-STAT5/STAT5. However, detection of the Ph-STAT5 band appears 'cleaner'. Presuming that the same lot of STAT5 (produced by in vitro transcription/translation) was used for all 4 samples in a specific experiment, the total amount of STAT5 should be the same for all 4 samples. Thus, if all 4 samples were detected on the same Ph-STAT5 blot, the intensities of the Ph-STAT5 bands should be directly comparable. If so, I suggest that the authors include the ratio of signals for [Ph-STAT5+SHP2]/[Ph-STAT5-] for the samples with and without HSP27. This ratio might show a more reproducible effect of HSP27 on dephosphorylation.

- Our answer:

Indeed, we confirm that in the figure 4b the same lot and amount of STAT5 produced in reticulocytes lysate was added in the assay (as in each of the other independent experiments). Thus, as requested by the reviewer, we included a bar graph representing the ratio of signal [Ph-STAT5+SHP2]/[Ph-STAT5] for the samples, with and without HSP27. Even though the error bars from the bar graph which represents the mean of 4 independent experiments does not change, the p value switch from $P < .05$ to $P < .01$ (New Fig.4b).

Other minor issues of clarity

2. **Regarding Fig. 3,** I did not realize before that the 2 actin rows probably indicate controls for 2 distinct blots. If so, it appears that the top 2 rows of samples (above the 1st actin row) are from 1 experiment and the 4 rows of samples above the bottom actin row are from a 2nd experiment. If this is correct, the authors could make this clearer by adding a dotted line below the 1st actin row, or by adding brackets to 1 side to show the grouping of samples from each experiment.

- Our answer:

As requested by the reviewer, we added a line below the 1st actin row to show the grouping of samples for each experiment (Fig. 3b)

3. Fig. 4 legend (pg 21, lines 640-641). The text is somewhat confusing. I presume the bars represent a normalized ratio of Ph-STAT5/STAT5 from the blots above. However, since they also show blots detecting SHP2rec, JAK2rec, and HSP27, the description on line 641 should be more explicit. Suggested text: "Bar graphs show normalized ratios of Ph-STAT5/STAT5 bands quantified from the western blots ..."

- Our answer:

We have now completed the legend Fig.4b to make it clearer for the reader.

4. Supplementary Fig. 4b and legend. Labels for the sample lanes in Supplementary Fig. 4b include the designations C2 and C1 (2nd and 3rd lanes, respectively). I presume these indicate 2 distinct sample concentrations loaded, but this should be noted in the figure legend.

- Our answer:

As presumed by the reviewer, the samples C1 and C2 correspond to 2 distinct volumes of HSP27 added in the reaction. We have now precised the designations C1 and C2 in the new supplementary legend Fig.5b.